# RényiCL: Contrastive Representation Learning with Skew Rényi Divergence

**Kyungmin Lee**    **Jinwoo Shin**
Korea Advanced Institute of Science and Technology (KAIST)
{kyungmnlee, jinwoos}@kaist.ac.kr

## Abstract

Contrastive representation learning seeks to acquire useful representations by estimating the shared information between multiple views of data. Here, the choice of data augmentation is sensitive to the quality of learned representations: as harder the data augmentations are applied, the views share more task-relevant information, but also task-irrelevant one that can hinder the generalization capability of representation. Motivated by this, we present a new robust contrastive learning scheme, coined RényiCL, which can effectively manage harder augmentations by utilizing Rényi divergence. Our method is built upon the variational lower bound of Rényi divergence, but a naïve usage of a variational method is impractical due to the large variance. To tackle this challenge, we propose a novel contrastive objective that conducts variational estimation of a skew Rényi divergence and provide a theoretical guarantee on how variational estimation of skew divergence leads to stable training. We show that Rényi contrastive learning objectives perform innate hard negative sampling and easy positive sampling simultaneously so that it can selectively learn useful features and ignore nuisance features. Through experiments on ImageNet, we show that Rényi contrastive learning with stronger augmentations outperforms other self-supervised methods without extra regularization or computational overhead. Moreover, we also validate our method on other domains such as graph and tabular, showing empirical gain over other contrastive methods. The implementation and pre-trained models are available at [1].

## 1 Introduction

Recently, many AI studies are enamored by the power of contrastive methods in learning useful representations without any human supervision, e.g., image [1, 2, 3, 4, 5, 6], text [7, 8], audio [9, 10] and multimodal video [11, 12, 13]. The key components of contrastive learning can be summarized into two-fold: data augmentation and contrastive objectives. The data augmentation generates different views of data where the views share information that is relevant to the downstream tasks. The contrastive objective then enforces the representation to capture the shared information between the views by bringing the views from the same data together and pushing the views from different data away in the representation space. Therefore, the choice of data augmentation is critical in contrastive learning, cf., see [3, 14, 15]. If the views share insufficient information, the representations cannot learn sufficient features for downstream tasks. On the other hand, if there is too much information between the views, the views might share nuisance features that hamper the generalization.

In this paper, we propose a new contrastive learning objective using Rényi divergence [16], coined *RényiCL*, which can effectively handle the case when the views share excessive information. The Rényi divergence is a generalization of KL divergence that is defined with an additional parameter

---

[1]https://github.com/kyungmnlee/RenyiCL

called an order. As the order becomes higher, the Rényi divergence penalizes more when two distributions mismatch. Therefore, since the goal of contrastive representation learning is to train an encoder to discriminate between positives and negatives, we hypothesize that maximizing the Rényi divergence between positives and negatives could lead to more discriminative representations when the data augmentations are aggressive. Furthermore, our theoretical analysis shows that RényiCL weighs importance on easy positives and hard negatives by using a higher order.

To implement RényiCL, one can consider variational lower bound of Rényi divergence [17]. However, we show that a variational lower bound of Rényi divergence does not suffice for contrastive learning due to their large variance. To tackle this challenge, we first made a key observation that the existing contrastive objectives are a variational form of a skew KL divergence, and show that variational estimation of skew divergence induces low variance property. Inspired by this, we consider a variational lower bound of a skew Rényi divergence and use it to implement RényiCL.

Through experiments under ImageNet [18], RényiCL achieves 76.2% linear evaluation accuracy, outperforming other self-supervised learning methods even using significantly less training epochs. Moreover, we show that RényiCL representations transfer to various datasets and tasks such as fine-grained object classification and few-shot classification, outperforming other self-supervised baselines. Finally, we validate the effectiveness of RényiCL on various domains such as vision, graph, and tabular datasets by showing empirical gain over conventional contrastive learning methods.

## 2 Related Works

Since Chen et al. [3] emphasized that the usage of data augmentation is crucial for contrastive learning, a series of works proposed various contrastive learning frameworks that showed great empirical success on visual representation learning [2, 3, 4, 5, 6, 19]. Inspired by the empirical success, recent works focused on optimal view generation for both positive and negatives. Tian et al. [14] proposed the InfoMin principle for view generation, which states that the views should share minimal information that is relevant to the downstream tasks. Another line of research focuses on hard negative samples [20, 21, 22] to enhance the performance of contrastive learning. Especially, Robinson et al. [20] proposed to use importance sampling to learn with hard negative samples.

Meanwhile, others are interested in searching for different contrastive objectives by modeling with various probabilistic divergences between positives and negatives. Oord et al. [23] proposed Contrastive Predictive Coding (CPC) (as known as InfoNCE) that is widely used in various contrastive learning scheme such as [2, 3, 4]. They theoretically prove that CPC objective is a variational lower bound to the mutual information, later [24] proposed Multi-Label CPC (MLCPC) which is a tighter lower bound to the mutual information. Hjelm et al. [25] proposed DeepInfoMax which used Jensen-Shannon divergence for variational estimation and maximization of mutual information. Zbontar et al. [26] proposed Barlow-Twins, which is shown to be a Hilbert-Schmidt Independence Criterion that approximates a Maximum Mean discrepancy measure between positives and negatives [27]. Ozair et al. [28] proposed Wasserstein Predictive Coding, which is both lower bound to the Wasserstein distance and mutual information. Similar to our approach, Tsai et al. [29] proposed relative predictive coding that uses skew $\chi^2$-divergence for contrastive learning, and empirically and theoretically show that variational estimation of skew $\chi^2$-divergence leads to stable contrastive representation learning. Lastly, our work is similar to [30], which proposed robust contrastive learning objective that uses symmetric binary classification loss for contrastive learning so that it can deal with noisy views.

## 3 Preliminaries

### 3.1 Rényi divergence

We first formally introduce Rényi divergence, which is a family of probability divergences including the popular Kullback-Leibler (KL) divergence as a special case. Let $P$, $Q$ be two probability distributions such that $P$ is absolutely continuous with respect to $Q$, denoted by $P \ll Q$ (i.e., the Radon-Nikodym derivative $dP/dQ$ exists). Then, the *Rényi divergence* [16, 31] of order $\gamma \in (0, 1) \cup (1, \infty)$ is defined by

$$R_\gamma(P \parallel Q) \coloneqq \frac{1}{\gamma(\gamma - 1)} \log \mathbb{E}_P\left[\left(\frac{dP}{dQ}\right)^{\gamma - 1}\right].$$

It is intertwined with various $f$-divergences, e.g., Rényi divergence with $\gamma \leftarrow 1$ and $\gamma = 2$ becomes KL divergence $D_{\text{KL}}(P \,\|\, Q)$ and a monotonic transformation of $\chi^2$ divergence, respectively. Remark that the Rényi divergence with higher order $\gamma$, penalizes more when the probability mass of $P$ does not overlap with $Q$ [32]. Rényi divergence has been studied in various machine learning tasks such as variational inference [33, 34] and training [35, 36] or evaluation of generative models [37].

## 3.2 Variational lower bounds of mutual information

The estimation and optimization of *mutual information* is an important topic in various machine learning tasks including representation learning [38]. The mutual information is defined by the KL divergence between the joint distribution and the product of marginal distributions. Formally, given two random variables $X \sim P_X$ and $Y \sim P_Y$, let $P_{XY}$ be the joint distribution of $X \times Y$. Then, we call the pair of samples $(x, y) \sim P_{XY}$ as a *positive pair*, and $(x, y) \sim P_X P_Y$ as a *negative pair*. Then, the mutual information is defined by KL divergence between positive pairs and negative pairs:

$$\mathcal{I}(X; Y) \coloneqq D_{\text{KL}}(P_{XY} \,\|\, P_X P_Y).$$

In general, the mutual information is intractable to compute unless the densities of $X$ and $Y$ are explicitly known, thus many works resort to optimize its variational lower bounds [38] associated with neural networks [39]. The Donsker-Varadhan (DV) [40] objective is a variational form of KL divergence defined as follows:

$$\mathcal{I}_{\text{DV}}(f) \coloneqq \mathbb{E}_P[f] - \log \mathbb{E}_Q[e^f], \quad \text{where} \quad D_{\text{KL}}(P \,\|\, Q) = \sup_{f \in \mathcal{F}} \mathcal{I}_{\text{DV}}(f), \tag{1}$$

where $\mathcal{F}$ is a set of bounded measurable functions on the support of $P$ and $Q$, and the optimum $f^* \in \mathcal{F}$ satisfies $f^* \propto \log \frac{dP}{dQ}$. Then, Belghazi et al. [39] proposed MINE that uses (1) as follows:

$$\mathcal{I}_{\text{MINE}}(f) = \mathbb{E}_{P_{XY}}[f(x, y)] - \log \mathbb{E}_{P_X P_Y}[e^{f(x,y)}].$$

However, Song et al. [41] showed that estimation with MINE objective might occur large variance unless one uses large number of samples, and Tsai et al. [29] also empirically evidenced that contrastive learning with the DV objective suffers from training instability.

To address the issue, the *contrastive predictive coding* (CPC) objective (also as known as In-foNCE) [23] is a popular choice for various practices including contrastive representation learning [2, 4, 3]. Given $B$ batch of samples $\{x_i\}_{i=1}^B$ from $X$, assume we have a single positive $y_i^+$ and $K$ negatives $\{y_{ij}^-\}_{j=1}^K$ for each $x_i$. Then, the CPC objective is defined as follows:

$$\mathcal{I}_{\text{CPC}}(f) \coloneqq \mathbb{E}_{(x, y^+) \sim P_{XY}, \{y_j^-\}_{j=1}^K \sim P_Y} \left[ \log \frac{(K+1) \cdot e^{f(x, y^+)}}{e^{f(x, y^+)} + \sum_{j=1}^K e^{f(x, y_j^-)}} \right],$$

where it is known that $\mathcal{I}_{\text{CPC}}(f) \leq \mathcal{I}(X; Y)$ for any bounded measurable function $f$ [23]. However, as $\mathcal{I}_{\text{CPC}}(f) \leq \log(K+1)$ for any $f$, the CPC objective becomes a high-bias estimator if the true value $\mathcal{I}(X; Y)$ is larger than $\log(K+1)$. To address this, Poole et al. [38] proposed $\alpha$-CPC which controls the bias by inserting $\alpha \in [0, 1]$ into CPC as follows:

$$\mathcal{I}_{\text{CPC}}^{(\alpha)}(f) \coloneqq \mathbb{E}_{(x, y^+) \sim P_{XY}, \{y_j^-\}_{j=1}^K \sim P_Y} \left[ \log \frac{e^{f(x, y^+)}}{\alpha e^{f(x, y^+)} + \frac{1-\alpha}{K} \sum_{j=1}^K e^{f(x, y_j^-)}} \right], \tag{2}$$

The $\alpha$-CPC objective also admits a variational lower bound to the mutual information $\mathcal{I}(X; Y)$ for any $\alpha \in [0, 1]$ (recovers the original CPC when $\alpha = \frac{1}{K+1}$), and it achieves smaller bias when using smaller value of $\alpha$. Furthermore, Song et al. [24] proposed $\alpha$-*Multi Label CPC* (MLCPC) which provides even a tighter lower bound to the mutual information. While CPC can be considered as $(K+1)$-way 1-shot classification problem, given $B$ batch of anchors $\{x_i\}_{i=1}^B$, MLCPC is equivalent to $B(K+1)$-way $B$-shot multi-label classification problem as follows:[2]

$$\mathcal{I}_{\text{MLCPC}}^{(\alpha)}(f) = \mathbb{E}_{\substack{\{(x_i, y_i^+)\}_{i=1}^B \sim P_{XY} \\ \{y_{ij}^-\}_{j=1}^K \sim P_Y}} \left[ \frac{1}{B} \sum_{i=1}^B \log \frac{e^{f(x_i, y_i^+)}}{\frac{\alpha}{B} \sum_{i=1}^B e^{f(x_i, y_i^+)} + \frac{1-\alpha}{BK} \sum_{i=1}^B \sum_{j=1}^K e^{f(x_i, y_{ij}^-)}} \right]. \tag{3}$$

---

[2]Remark that we use slightly different form as in [24] to scale $\alpha$ to be reside in $[0, 1)$.

# 4 Variational Estimation of Skew Rényi Divergence

Similar to variational estimators of $D_{\text{KL}}(P_{XY} \| P_X P_Y)$ in the previous section, we consider a variational estimator of Rényi divergence $R_\gamma(P_{XY} \| P_X P_Y)$, in particular, considering its application to contrastive representation learning. To that end, we first introduce the following known lemma which states variational representation of Rényi divergence similar to the DV objective in (1).

**Lemma 4.1** ([17]). *For distributions $P, Q$ such that $P \ll Q$, let $\mathcal{F}$ be a set of bounded measurable functions. Then Rényi divergence of order $\gamma \in (0,1) \cup (1,\infty)$ admits following variational form:*

$$R_\gamma(P \| Q) = \sup_{f \in \mathcal{F}} \mathcal{I}_{\text{Renyi}}^{(\gamma)}(f) \quad for \quad \mathcal{I}_{\text{Renyi}}^{(\gamma)}(f) := \frac{1}{\gamma - 1} \log \mathbb{E}_P[e^{(\gamma-1)f}] - \frac{1}{\gamma} \log \mathbb{E}_Q[e^{\gamma f}], \quad (4)$$

*where the optimum $f^* \in \mathcal{F}$ satisfies $f^* \propto \log \frac{dP}{dQ}$. Also, as $\gamma \to 1$, (4) becomes DV objective in (1).*

## 4.1 Challenges in variational Rényi divergence estimation

To perform contrastive representation learning, one can use the variational estimator (4) for $R_\gamma(P_{XY} \| P_X P_Y)$, similarly as did for KL divergence. However, the contrastive learning with (4) is impractical due to the large variance. In Appendix C.2, we show that even for a simple synthetic Gaussian dataset, (4) is not available due to exploding variance when estimating Rényi divergence between two highly-correlated Gaussian distributions. This exploding-variance issue of (4) resembles that of the variational mutual information estimator well observed in the literature [29, 38, 41]. In the following theorem, we provide an analogous result for the Rényi variational objective (4).

**Theorem 4.1.** *Assume $P \ll Q$, and $\text{Var}_Q[dP/dQ] < \infty$. Let $P_m$ and $Q_n$ be the empirical distributions of $m$ i.i.d samples from $P$ and $n$ i.i.d samples from $Q$, respectively. Define*

$$\hat{\mathcal{I}}_{\text{Renyi}}^{(\gamma)}(f) := \frac{1}{\gamma - 1} \log \mathbb{E}_{P_m}[e^{(\gamma-1)f}] - \frac{1}{\gamma} \log \mathbb{E}_{Q_n}[e^{\gamma f}],$$

*and assume we have $f^* \propto \log(dP/dQ)$. Then $\forall \gamma > 1, \forall m \in \mathbb{N}$, we have*

$$\lim_{n \to \infty} n \cdot \text{Var}_{P,Q}[\hat{\mathcal{I}}_{\text{Renyi}}^{(\gamma)}(f^*)] \geq \frac{e^{\gamma^2 D_{\text{KL}}(P \| Q)} - \gamma^2}{e^{2\gamma(\gamma-1)R_\gamma(P \| Q)}}.$$

The proof of Theorem 4.1 is in Appendix A.1. Theorem 4.1 implies that even though one achieves the optimal function for variational estimation, the variance of (4) could explode exponentially with respect to the ground-truth mutual information. This result is coherent with [42], which identified the problems in variational estimation of mutual information.

On the other hand, we recall that CPC and MLCPC are empirically shown to be low variance estimators [38, 24]. Then, the natural question arises: what makes CPC and MLCPC fundamentally different from the DV objective? In the following section, we answer to this question by showing that the CPC and MLCPC objectives are variational lower bounds of a skew KL divergence, and provide a theoretical evidence that variational estimators of skew divergence can have low variance. This insight will be used later for designing a low-variance estimator for the desired Rényi divergence.

## 4.2 Contrastive learning objectives are variational skew-divergence estimators

We first introduce the definition of $\alpha$-*skew KL divergence*. For distributions $P, Q$ with $P \ll Q$ and for any $\alpha \in [0,1]$, the $\alpha$-skew KL divergence between $P$ and $Q$ is defined by the KL divergence between $P$ and the mixture $\alpha P + (1-\alpha)Q$:

$$D_{\text{KL}}^{(\alpha)}(P \| Q) := D_{\text{KL}}(P \| \alpha P + (1-\alpha)Q).$$

One can see that the DV objective (1) for $\alpha$-skew KL divergence can be written as following:

$$D_{\text{KL}}^{(\alpha)}(P \| Q) = \sup_{f \in \mathcal{F}} \mathbb{E}_P[f] - \log \left( \alpha \mathbb{E}_P[e^f] + (1-\alpha)\mathbb{E}_Q[e^f] \right). \quad (5)$$

Then following theorem reveals that $\alpha$-CPC and $\alpha$-MLCPC are variational lower bounds of $\alpha$-skew divergence between $P_{XY}$ and $P_X P_Y$, $D^{(\alpha)}(P_{XY} \| P_X P_Y)$

**Theorem 4.2.** *For any $\alpha \in (0, 1/2)$, the $\alpha$-CPC and $\alpha$-MLCPC are variational lower bound of $\alpha$-skew KL divergence between $P_{XY}$ and $P_X P_Y$, i.e., the following holds:*

$$\sup_{f \in \mathcal{F}} \mathcal{I}_{\mathrm{MLCPC}}^{(\alpha)}(f) = \sup_{f \in \mathcal{F}} \mathcal{I}_{\mathrm{CPC}}^{(\alpha)}(f) = D_{\mathrm{KL}}^{(\alpha)}(P_{XY} \parallel P_X P_Y)$$

Here we provide a simple proof sketch and the full proof is in Appendix A.2. From the Jensen's inequality, one can see that the variational form in (5) is a lower bound to the $\alpha$-MLCPC. Thus, we have $D_{\mathrm{KL}}^{(\alpha)}(P_{XY} \parallel P_X P_Y) \leq \sup_f \mathcal{I}_{\mathrm{MLCPC}}^{(\alpha)}(f)$. On the other hand, we also show that $\mathcal{I}_{\mathrm{MLCPC}}^{(\alpha)}(f) \leq D_{\mathrm{KL}}^{(\alpha)}(P_{XY} \parallel P_X P_Y)$ for all $f \in \mathcal{F}$. Therefore, we show that $\alpha$-MLCPC is a variational lower bound of $D_{\mathrm{KL}}^{(\alpha)}(P_{XY} \parallel P_X P_Y)$. Note that the same argument holds for $\alpha$-CPC. Remark that Theorem 4.2 implies that $\alpha$-CPC and $\alpha$-MLCPC are strictly loose bounds of mutual information. In particular, from the convexity of KL divergence, the following holds for any $\alpha > 0$:

$$D_{\mathrm{KL}}^{(\alpha)}(P_{XY} \parallel P_X P_Y) \leq (1 - \alpha) D_{\mathrm{KL}}(P_{XY} \parallel P_X P_Y) < D_{\mathrm{KL}}(P_{XY} \parallel P_X P_Y).$$

Also, Theorem 4.2 reveals that $\alpha$-CPC and $\alpha$-MLCPC can be written as following population form:

$$\mathcal{I}_{\mathrm{CPC}}^{(\alpha)}(f) = \mathbb{E}_{P_{XY}}[f(x, y)] - \mathbb{E}_{P_X}\left[\log\left(\alpha \mathbb{E}_{P_{Y|X}}[e^{f(x,y)}] + (1 - \alpha)\mathbb{E}_{P_Y}[e^{f(x,y)}]\right)\right]$$

$$\mathcal{I}_{\mathrm{MLCPC}}^{(\alpha)}(f) = \mathbb{E}_{P_{XY}}[f(x, y)] - \log\left(\alpha \mathbb{E}_{P_{XY}}[e^{f(x,y)}] + (1 - \alpha)\mathbb{E}_{P_X P_Y}[e^{f(x,y)}]\right).$$

**Variational lower bounds of skew-divergence have low variance** Further, we formally show that the variational estimation of empirical skew divergence has low variance, explaining why CPC and MLCPC objectives become low-variance estimators of the mutual information. The following theorem shows that the variance of the variational estimator can be adjusted with the right choice of $\alpha$.

**Theorem 4.3.** *Assume $P \ll Q$ and $Var_Q[dP/dQ] < \infty$. Let $P_m$ and $Q_n$ be empirical distributions of $m$ i.i.d samples from $P$ and $n$ i.i.d samples from $Q$. Then define*

$$\hat{\mathcal{I}}_{\mathrm{KL}}^{(\alpha)}(f) = \mathbb{E}_{P_m}[f] - \log\left(\alpha \mathbb{E}_{P_m}[e^f] + (1 - \alpha)\mathbb{E}_{Q_n}[e^f]\right),$$

*and assume that there is $\hat{f} \in \mathcal{F}$ that $|\mathcal{I}_{\mathrm{KL}}^{(\alpha)}(\hat{f}) - D_{\mathrm{KL}}^{(\alpha)}(P_m \parallel Q_n)| < \varepsilon_f$ for some $\varepsilon_f > 0$. Then for $\forall \alpha < 1/8$, the variance of estimator satisfies*

$$Var_{P,Q}\left[\hat{\mathcal{I}}_{\mathrm{KL}}^{(\alpha)}(\hat{f})\right] \leq c_1 \varepsilon_f + \frac{c_2(\alpha)}{\min\{n, m\}} + \frac{c_3 \log^2(\alpha m)}{m} + \frac{c_4 \log^2(c_5 n)}{\alpha^2 n},$$

*for some constants $c_1, c_3, c_4, c_5 > 0$ that are independent of $n$, $m$, $\alpha$, and $D_{\mathrm{KL}}(P\|Q)$, and $c_2(\alpha)$ satisfies $c_2(\alpha) = \min\{\frac{1}{\alpha}, \frac{\chi^2(P\|Q)}{1-\alpha}\}$, where $\chi^2(P\|Q) = \mathbb{E}_Q[(dP/dQ)^2]$.*

The proof is in Appendix A.3. Since we have $\chi^2(P\|Q) \geq e^{D_{\mathrm{KL}}(P\|Q)} - 1$, if $\alpha$ is too small, the bound in Theorem 4.3 is loose as in Theorem 4.1. Therefore, Theorem 4.3 implies that one should use sufficiently large $\alpha$ to achieve low variance. Theorem. 4.3 demonstrates that if one can find a sufficiently close critic for empirical skew KL divergence, the variance of the estimator is asymptotically bounded unless we choose large enough $\alpha = \alpha_n = \omega(n^{-1/2})$.

## 4.3 Variational estimation of skew Rényi divergence

Inspired by the theoretical guarantee that skew divergence can achieve bounded variance, we now present a variational estimator of skew Rényi divergence. Remark that the analogous version of Theorem 4.3 can be achieved for Rényi divergence (see Appendix A.4). For any $\alpha \in [0, 1]$ and $\gamma \in (0, 1) \cup (1, \infty)$, the $\alpha$-skew Rényi divergence of order $\gamma$ is defined as:

$$R_\gamma^{(\alpha)}(P \parallel Q) \coloneqq R_\gamma(P \parallel \alpha P + (1 - \alpha)Q).$$

Then, we define $(\alpha, \gamma)$-*Rényi Multi-Label CPC* (RMLCPC) objective by using the variational lower bound (4) for $\alpha$-skew Rényi divergence between $P_{XY}$ and $P_X P_Y$:

$$\mathcal{I}_{\mathrm{RMLCPC}}^{(\alpha, \gamma)}(f) \coloneqq \frac{1}{\gamma - 1} \log \mathbb{E}_{P_{XY}}[e^{(\gamma - 1)f(x,y)}] - \frac{1}{\gamma} \log(\alpha \mathbb{E}_{P_{XY}}[e^{\gamma f(x,y)}] + (1 - \alpha)\mathbb{E}_{P_X P_Y}[e^{\gamma f(x,y)}]),$$

which satisfies $\sup_f \mathcal{I}_{\text{RMLCPC}}^{(\alpha,\gamma)}(f) = R_\gamma^{(\alpha)}(P_{XY} \| P_X P_Y)$ for any $\alpha \in [0,1]$ and $\gamma \in (0,1) \cup (1,\infty)$.

Note that the RMLCPC objective can be used for contrastive learning with multiple positive views. For example, when using multi-crops data augmentation [43], instead of computing contrastive objective for each crop, we gather all positive pairs and negative pairs and compute only once for the final loss. The pseudo-code for the RMLCPC objective is in Appendix Algorithm 1.

# 5   Rényi Contrastive Representation Learning

In this section, we present *Rényi contrastive learning* (RényiCL) where we use the RMLCPC objective for contrastive representation learning. As we discussed earlier, the Rényi divergence penalizes more when two distributions differ. Thus, when using harder data augmentations in contrastive learning, one can expect that RényiCL can learn more discriminative representation.

## 5.1   Rényi contrastive representation learning

We begin with backgrounds for contrastive representation learning. Given a dataset $\mathcal{X}$, the goal of representation learning is to train an encoder $g : \mathcal{X} \to \mathbb{R}^d$ that is a useful feature of $\mathcal{X}$. Especially, contrastive representation learning enforces $g$ to discriminate between positive pairs and negative pairs [2], where positive pairs are generated by applying data augmentation on the same data and negative pairs are augmented data from different source data. Formally, let $V, V'$ be random variables for augmented views from dataset $\mathcal{X}$. Then denote $P_{VV'}$ be the joint distribution of a positive pair and $P_V P_{V'}$ be the distribution of negative pairs. For the sake of brevity, we denote $(z, z^+) \sim P_{VV'}$ be a positive pair, and $(z, z^-) \sim P_V P_{V'}$ be a negative pair. Also, let $z = g(v), z' = g(v')$ be features of the encoder and $Z = g(V), Z' = g(V')$ be corresponding random variables of feature distributions. Then we define $(z, z^+) \sim P_{ZZ'}$ be features of positive views, and $(z, z^-) \sim P_Z P_{Z'}$ be features of negative views.

The InfoMax principle [25] for representation learning aims to find $g$ that preserves the maximal mutual information between $V$ and $V'$ by following:

$$\sup_{g:\mathcal{X}\to\mathbb{R}^d} D_{\text{KL}}(P_{ZZ'} \| P_Z P_{Z'}) = \sup_{g:\mathcal{X}\to\mathbb{R}^d} \mathcal{I}(Z; Z') \leq \mathcal{I}(V; V'),$$

i.e., finds a neural encoder $g$ which discriminates between positive and negative pairs as much as possible. Here, $\alpha$-CPC or $\alpha$-MLCPC are used for plug-in variational estimators of mutual information, for example contrastive learning with $\alpha$-MLCPC satisfies following:

$$\sup_{g:\mathcal{X}\to\mathbb{R}^d} \sup_{f\in\mathcal{F}} \mathcal{I}_{\text{MLCPC}}^{(\alpha)}(f,g) = \sup_{g:\mathcal{X}\to\mathbb{R}^d} D_{\text{KL}}^{(\alpha)}(P_{ZZ'} \| P_Z P_{Z'}),$$

since $\mathcal{I}_{\text{MLCPC}}^{(\alpha)}(f,g)$ is a variational estimator of $D_{\text{KL}}^{(\alpha)}(P_{ZZ'} \| P_Z P_{Z'})$.

Now we present Rényi contrastive representation learning, which considers the following optimization problem by using skew Rényi divergence:

$$\sup_{g:\mathcal{X}\to\mathbb{R}^d} \sup_{f\in\mathcal{F}} \mathcal{I}_{\text{RMLCPC}}^{(\alpha,\gamma)}(f,g) = \sup_{g:\mathcal{X}\to\mathbb{R}^d} R_\gamma^{(\alpha)}(P_{ZZ'} \| P_Z P_{Z'}),$$

since $\mathcal{I}_{\text{RMLCPC}}^{(\alpha,\gamma)}(f,g)$ is a variational estimator of $R_\gamma^{(\alpha)}(P_{ZZ'} \| P_Z P_{Z'})$.

## 5.2   Gradient analysis for RényiCL

In this section, we provide an analysis on the effect of $\gamma$ in RényiCL based on the gradient of contrastive objectives. For simplicity, we consider the case when $\alpha = 0$, and we provide general analysis for $\alpha > 0$ in Appendix B.1. Let $f_\theta$ be a neural network parameterized by $\theta$. By using the reparametrization trick, the gradient of $\mathcal{I}_{\text{MLCPC}}(f_\theta)$ becomes

$$\nabla_\theta \mathcal{I}_{\text{MLCPC}}(f_\theta) = \mathbb{E}_{z,z^+}[\nabla_\theta f_\theta(z, z^+)] - \frac{\mathbb{E}_{z,z^-}[e^{f_\theta(z,z^-)}\nabla_\theta f_\theta(z, z^-)]}{\mathbb{E}_{z,z^-}[e^{f_\theta(z,z^-)}]}$$

$$= \mathbb{E}_{z,z^+}[\nabla_\theta f_\theta(z, z^+)] - \mathbb{E}_{\text{sg}(q_\theta(z,z^-))}[\nabla_\theta f_\theta(z, z^-)],$$

Table 1: Linear evaluation on the ImageNet validation set. We report pre-training epochs and Top-1 classification accuracies (%).

| Method | Epochs | Top-1 |
|---|---|---|
| SimCLR [3] | 800 | 70.4 |
| Barlow Twins [26] | 800 | 73.2 |
| BYOL [47] | 800 | 74.3 |
| MoCo v3 [6] | 800 | 74.6 |
| SwAV [43] | 800 | 75.3 |
| DINO [43] | 800 | 75.3 |
| NNCLR [19] | 1000 | 75.6 |
| C-BYOL [48] | 1000 | 75.6 |
| **RényiCL** | **200** | **75.3** |
| **RényiCL** | **300** | **76.2** |

Table 2: Semi-supervised learning results on ImageNet. We report Top-1 and Top-5 classification accuracies (%) by fine-tuning a pre-trained ResNet-50 with 1% and 10% ImageNet datasets.

| | 1% ImageNet | | 10% ImageNet | |
|---|---|---|---|---|
| Method | Top-1 | Top-5 | Top-1 | Top-5 |
| Supervised [3] | 25.4 | 48.4 | 56.4 | 80.4 |
| SimCLR [3] | 48.3 | 75.5 | 65.6 | 87.8 |
| BYOL [47] | 53.2 | 78.4 | 68.8 | 89.0 |
| SwAV [43] | 53.9 | 78.5 | 70.2 | 89.9 |
| Barlow Twins [26] | 55.0 | 79.2 | 69.7 | 89.3 |
| NNCLR [19] | 56.4 | 80.7 | 69.8 | 89.3 |
| C-BYOL [48] | **60.6** | **83.4** | 70.5 | 90.0 |
| **RényiCL** | 56.4 | 80.6 | **71.2** | **90.3** |

where $q_\theta(z, z^-) \propto e^{f_\theta(z,z^-)}$ is a self-normalized importance weights [44], and sg is a stop-gradient operator. Thus, the MLCPC objective is equivalent to following in terms of gradient [45]:

$$\mathcal{I}_{\text{MLCPC}}(f_\theta) = \mathbb{E}_{z,z^+}[f_\theta(z, z^+)] - \mathbb{E}_{\text{sg}(q_\theta(z,z^-))}[f_\theta(z, z^-)]. \tag{6}$$

This shows that the original contrastive objective performs innate hard negative sampling with importance weight $q_\theta$ [20], as the gradient of negative pairs becomes larger as the value of $f_\theta$ becomes larger. On the other hand, for RMLCPC objective, by letting $\alpha = 0$, and taking gradient with respect to $\theta$, we have

$$\nabla_\theta \mathcal{I}_{\text{RMLCPC}}^{(\gamma)}(f_\theta) = \frac{\mathbb{E}_{z,z^+}[e^{(\gamma-1)f_\theta(z,z^+)}\nabla_\theta f_\theta(z, z^+)]}{\mathbb{E}_{z,z^+}[e^{(\gamma-1)f_\theta(z,z^+)}]} - \frac{\mathbb{E}_{z,z^-}[e^{\gamma f_\theta(z,z^-)}\nabla_\theta f_\theta(z, z^-)]}{\mathbb{E}_{z,z^-}[e^{\gamma f_\theta(z,z^-)}]}$$

$$= \mathbb{E}_{\text{sg}(q_\theta(z,z^+;\gamma-1))}[\nabla_\theta f_\theta(z, z^+)] - \mathbb{E}_{\text{sg}(q_\theta(z,z^-;\gamma))}[\nabla_\theta f_\theta(z, z^-)],$$

where $\text{sg}(q_\theta(z, z^+; \gamma-1)) \propto e^{(\gamma-1)f_\theta(z,z^+)}$ and $\text{sg}(q_\theta(z, z^-; \gamma)) \propto e^{\gamma f_\theta(z,z^-)}$ are self-normalizing importance weights for each positive and negative term. Hence, the RMLCPC objective is equivalent to the following in terms of gradient:

$$\mathcal{I}_{\text{RMLCPC}}^{(\gamma)}(f_\theta) = \mathbb{E}_{\text{sg}(q_\theta(z,z^+;\gamma-1))}[f_\theta(z, z^+)] - \mathbb{E}_{\text{sg}(q_\theta(z,z^-;\gamma))}[f_\theta(z, z^-)]. \tag{7}$$

Remark that (6) and (7) have common form that $f_\theta$ is maximized for positive pairs and minimized for negative pairs, which is similar to contrastive divergence [46]. On the other hand, (6) and (7) have two differences:

- **Hard negative sampling**: the gradient weighs more on harder negatives, i.e., $(z, z^-)$ with high value of $f_\theta(z, z^-)$, as $\gamma$ increases [20].
- **Easy positive sampling**: the gradient weighs more on easier positives, i.e., $(z, z^+)$ with high value of $f_\theta(z, z^+)$ as $\gamma \in (1, \infty)$ increases.

Therefore, when there are positive views that share task-irrelevant information, the RMLCPC objective resists updating, and it regularizes the model to ignore task-irrelevant information. The hard negative sampling with importance weight was proposed in [20], but our analysis shows that CPC and MLCPC conduct intrinsic hard negative sampling. Also, the Rényi contrastive learning can control the level of hard negative sampling by choosing the appropriate $\gamma$. Lastly, remark that our analysis is given for $\alpha = 0$, but in practice we use nonzero value of $\alpha$. In Appendix B.1, we show that it requires sufficiently small values of $\alpha$ to have the effect of easy positive sampling for harder data augmentations.

## 6 Experiments

### 6.1 RényiCL for visual representation learning on ImageNet

**Setup.** For ImageNet [18] experiments, we use ResNet-50 [49] for encoder $g$. We use MLP projection head with momentum encoder [4], which is updated by EMA, and we use predictor, following the

Table 3: Transfer learning performance on object classification datasets. For Aircraft and Flowers, we report mean per-class accuracy (%), and for VOC2007, we report 11-point mAP. Otherwise, we report Top-1 classification accuracies (%).

| Method | CIFAR10 | CIFAR100 | Food101 | Flowers | Cars | Aircraft | DTD | SUN397 | VOC2007 |
|---|---|---|---|---|---|---|---|---|---|
| Supervised [3] | 93.6 | 78.3 | 72.3 | 94.7 | 66.7 | 61.0 | 74.9 | 61.9 | 82.8 |
| SimCLR [3] | 90.6 | 71.6 | 68.4 | 91.2 | 50.3 | 50.3 | 74.5 | 58.8 | 81.4 |
| BYOL [47] | 91.3 | 78.4 | 75.3 | 96.1 | 67.8 | 60.6 | 75.5 | 62.2 | 82.5 |
| NNCLR [19] | 93.7 | **79.0** | 76.7 | 95.1 | 67.1 | **64.1** | 75.5 | 62.5 | 83.0 |
| **RényiCL** | **94.4** | **79.0** | **78.0** | **96.5** | **71.5** | 61.8 | **77.3** | **66.1** | **88.2** |

Table 4: Comparison on contrastive self-supervised methods with harder augmentations. We compare Top-1 linear evaluation accuracy (%) on ImageNet, Pets, Caltech101 datasets and few-shot classification accuracy (%) over 2000 episodes on FC100, CUB200, and Plant disease datasets. $^\dagger$ denotes trained without multi-crops. Here, we consider RényiCL without multi-crops as other baselines do, i.e., for fair comparison.

| Method | Epochs | Linear evaluation | | | 5-way 1-shot | | | 5-way 5-shot | | |
|---|---|---|---|---|---|---|---|---|---|---|
| | | ImageNet | Pets | Caltech101 | FC100 | CUB200 | Plant | FC100 | CUB200 | Plant |
| InfoMin [14] | 800 | **73.0** | 86.37 | 88.48 | 31.80 | 53.81 | 66.11 | 45.09 | 72.20 | 84.12 |
| CLSA [59] | 800 | 72.2 | 85.18 | 91.21 | 34.46 | 50.83 | 67.39 | 49.16 | 67.93 | 86.15 |
| **RényiCL**$^\dagger$ | 200 | 72.6 | **88.43** | **94.04** | **36.31** | **59.73** | **80.68** | **53.39** | **82.12** | **94.29** |

practice of [47, 6]. Then the critic $f$ is implemented by the cosine similarity between the output of momentum encoder and base encoder with predictor, divided by temperature $\tau = 0.5$. We maximize RMLCPC objective with $\gamma = 2.0$.

**Data augmentation.** Given the popular data augmentation baseline [47, 3], we further use RandAugment [50] and multi-crop augmentation [43] to implement harder data augmentation. When using multi-crop, we gather all positives and negatives, then directly compute the RMLCPC objective, which differs from the original method in [43] (see 1 in Appendix B.1).

**Linear evaluation.** We follow the linear evaluation protocol, where we report the Top-1 ImageNet validation accuracy (%) of a linear classifier trained on the top of frozen features. Table 1 compares the performance of different self-supervised methods using linear evaluation protocol. First, RényiCL outperforms other self-supervised methods with a large margin by only training for 300 epochs. Meanwhile, RényiCL enjoys better computational efficiency in that it does not require a large batch size [3, 47, 43, 6] or extra memory queue [19].

**Semi-supervised learning.** We evaluate the usefulness of the learned feature in a semi-supervised setting with 1% and 10% subsets of ImageNet dataset [3, 43]. The results are in Table 2. We observe that RényiCL achieves the best performance on fine-tuning with 10% subset, and runner-up on fine-tuning with 1% subset, showing the generalization capacity in low-shot learning scenarios.

**Transfer learning.** We evaluate the learned representation through the performance of transfer learning on fine-grained object classification datasets by using linear evaluation. We use CIFAR10/100 [51], Food101 [52], Flowers [53], Cars [54], Aircraft [55], DTD [56], SUN397 [57], and VOC2007 [58]. In Table 3, we compare it with other self-supervised methods. Note that RényiCL achieves the best performance in 8 out of 9 datasets, especially showing superior performance on SUN397 (3.6%) and VOC2007 (5.2%) datasets.

**Comparison among self-supervised methods using hard augmentations.** To demonstrate the effectiveness of RényiCL on learning with harder augmentations, we compare with InfoMin [14] and CLSA [59] where they also use harder augmentations for contrastive representation learning. Table 4 demonstrates the performance of representations learned with different self-supervised methods by linear evaulation on ImageNet [18], Pets [60], Caltech101 [61], and few-shot classification on FC100 [62], Caltech-UCSD Birds (CUB200) [50], and Plant Disease [63] datasets. While RényiCL achieves slightly lower performance than InfoMin on ImageNet validation accuracy, it outperforms InfoMin and CLSA on other transfer learning tasks with a large margin, which indicates the superiority of RényiCL on learning better representations under the hard data augmentations.

Table 5: Ablation on contrastive objectives for unsupervised representation learning on image (ImageNet-100, CIFAR-100, CIFAR-10) and tabular (CovType, Higgs-100K) datasets. We report Top-1 accuracy (%) with linear evaluation. Average over 5 runs.

| | ImageNet-100 | | CIFAR-100 | | CIFAR-10 | | CovType | | Higgs-100K | |
| Method | Base | Hard | Base | Hard | Base | Hard | Base | Hard | Base | Hard |
|---|---|---|---|---|---|---|---|---|---|---|
| CPC | 79.7 | 81.1(+1.4) | 65.4 | 67.1(+1.7) | 91.7 | 91.9(+0.2) | 71.6 | 74.3(+2.7) | 64.7 | 71.3(+6.6) |
| MLCPC | 79.5 | 81.2(+1.7) | 65.6 | 66.6(+1.0) | 91.9 | 92.1(+0.2) | 71.7 | 74.1(+2.4) | 64.9 | 71.5(+6.6) |
| RMLCPC | 78.9 | **81.6(+2.7)** | 64.5 | **68.5(+4.0)** | 90.7 | **92.5(+1.8)** | 72.1 | **74.9(+2.8)** | 64.5 | **72.4(+7.9)** |

Table 6: Comparison of different contrastive learning methods on graph TUDataset. Average over 5 runs.

| Method | NCI1 | PROTEINS | DD | MUTAG | COLLAB | RDT-B | RDT-M5K | IMDB-B |
|---|---|---|---|---|---|---|---|---|
| InfoGraph [72] | 76.20 | 74.44 | 72.85 | 89.01 | 70.65 | 82.50 | 53.46 | **73.03** |
| GraphCL [15] | 77.87 | 74.39 | 78.62 | 86.80 | 71.36 | 89.53 | 55.99 | 71.14 |
| JOAO [73] | 78.07 | 74.55 | 77.32 | 87.35 | 69.50 | 85.29 | 55.74 | 70.21 |
| JOAOv2 [73] | 78.36 | 74.07 | 77.40 | 87.67 | 69.33 | 86.42 | 56.03 | 70.83 |
| **RényiCL** | **78.60** | **75.11** | **78.98** | **90.22** | **71.88** | **90.92** | **56.18** | 72.38 |

## 6.2 RényiCL for representation learning on various domains

**Setup.** We perform ablation studies of RényiCL on various domains such as images, tabular, and graph datasets. For images, we consider CIFAR-10/100 [51] and ImageNet-100 [64], which is a 100-class subset of ImageNet [18]. For the base data augmentation, we use a popular benchmark from [3], and for the harder augmentation, we further apply RandAugment [50] and RandomErasing [65]. For ImageNet-100, we use the default settings of [6] with ResNet-50 backbone with only change in learning objective. For CIFAR-10 and CIFAR-100, we follow the settings in [3] with ResNet-18 backbone adjusted for the CIFAR dataset. We use same $\alpha$ for all CPC, MLCPC, RMLCPC, and use $\gamma = 2.0$ for RMLCPC.

For tabular experiments, we use Forest Cover Type (CovType) and Higgs Boson (Higgs) [66] dataset from UCI repository [67]. Due to its massive size, we consider a subset of 100K for the Higgs experiments. Since the tabular dataset has limited domain knowledge, we do not use any data augmentation for the baseline and when using data augmentation, we use random masking noise [68] for the Higgs dataset and random feature corruption [69] for the CovType dataset. We follow the settings of MoCo v3 [6] except that a 5-layer MLP is used for the backbone. We use $\gamma = 1.1$ for CovType and $\gamma = 1.2$ for Higgs. For evaluation, we use linear evaluation for CovType and we compute the linear regression layer by pseudo-inverse between the feature matrix and label matrix.

Lastly, we examine RényiCL on graph TUDataset [70], which contains numerous graph machine learning benchmarks of different domains such as bioinformatics, molecules, and social network. We follow the experimental setup of [15]: we use graph isomorphism network [71] backbone and use node dropout, edge perturbation, attribute masking, and subgraph sampling for data augmentations. For the graph experiments, we do not change the strength of data augmentation nor add new data augmentation. The detailed hyper-parameters for training and evaluation for the whole dataset can be found in Appendix B.3.

**Results.** Table 5 compares the Top-1 linear evaluation accuracy (%) of representations with different contrastive learning objectives. We observe that RMLCPC obtains the best performance when using harder data augmentations on all images and tabular datasets. Also, RMLCPC has the largest gain on using harder augmentation, showing that the effectiveness of RényiCL is amplified when we use stronger augmentation. Table 6 compare the Top-1 linear evaluation accuracy (%) of graph RényiCL with other graph contrastive learning methods. Even without harder data augmentation, RényiCL achieves the best performance in 7 out of 8 benchmarks. Especially, RényiCL outperforms JOAO [73], where the views are learned to boost the performance of graph contrastive learning.

### 6.3 Mutual information estimation and view selection

Remark that CPC and MLCPC are intrinsically high-bias mutual information estimator as $\alpha$-skew KL divergence is strictly smaller than original KL divergence for $0 < \alpha < 1$, i.e.,

$$D_{\texttt{KL}}^{(\alpha)}(P\|Q) \le (1-\alpha)D_{\texttt{KL}}(P\|Q) < D_{\texttt{KL}}(P\|Q),$$

from the convexity of KL divergence. Similarly, RMLCPC is also a strictly-biased estimator of Rényi divergence. However, one can recover the original mutual information by using the optimal condition of function $f$ from all CPC, MLCPC, and RMLCPC objectives [29]. Here, we describe the sketch and present detailed method in Appendix C.1. From the optimality condition of (1) and (4), the optimal critic $f^*$ of $\alpha$-CPC, $\alpha$-MLCPC and $(\alpha,\gamma)$-RMLCPC of any $\gamma$ satisfies $f^* \propto \log \frac{dP_{XY}}{\alpha dP_{XY} + (1-\alpha)dP_X dP_Y}$, then we use Monte-Carlo method [44] to approximate log-normalization constant, and use it to approximate the log-density ratio $\hat{r} = \log \frac{dP_{XY}}{dP_X dP_Y}$. Then we estimate the mutual information by $\hat{\mathcal{I}}(X;Y) = \frac{1}{B}\sum_{i=1}^{B}\log\hat{r}(x_i, y_i)$, where $(x_i, y_i) \sim P_{XY}$ are positive pairs.

**Mutual information estimation of synthetic Gaussian.** Following [38, 24], we conduct experiments on estimating the mutual information of multivariate Gaussian distributions. In Appendix C.2, we show that while $\alpha$-CPC, $\alpha$-MLCPC, and $(\alpha,\gamma)$-RMLCPC objectives are high biased objectives, one can achieve low bias, low variance estimator by using the approximation method above.

## 7 Conclusion and Discussion

We propose Rényi Contrastive Learning (RényiCL) which utilizes Rényi divergence to deal with stronger data augmentations. Since the variational lower bound of Rényi divergence is insufficient for contrastive learning due to large variance, we introduce a variational lower bound of skew Rényi divergence, namely Rényi-MLCPC. Indeed, we show that CPC and MLCPC are variational forms of skew KL divergence, and provide theoretical analysis on how they achieve low variance. Through experiments, we validate the effectiveness of RényiCL by using harder data augmentations.

**Limitations and future works.** While RényiCL is beneficial when using harder data augmentations, we did not identified the optimal data augmentation strategy. One can use policy search method on data augmentation [74] with our RényiCL similar to that of [75]. Also, we think that the proposed scheme based on Rényi divergence could be useful for other tasks, e.g., information bottleneck [76]. Lastly, many works focused on the diverse perspectives of contrastive learning [45, 77, 78], where those approaches could be complementary to our work. We leave them for future works.

**Negative societal impacts.** Since contrastive learning often requires long epochs of training, it raises environmental concerns, e.g. carbon generation. Nevertheless, RényiCL is shown to be effective even under a smaller number of epochs, compared to existing schemes.

## Acknowledgement

This work was partly supported by Institute of Information & communications Technology Planning & Evaluation (IITP) grant funded by the Korea government(MSIT) (No.2019-0-00075, Artificial Intelligence Graduate School Program (KAIST)) and Institute of Information & communications Technology Planning & Evaluation (IITP) grant funded by the Korea government(MSIT) (No.2022-0-00959, Few-shot Learning of Causal Inference in Vision and Language for Decision Making).

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
