# A Proofs

## A.1 Proof of Theorem 4.1

*Proof.* Without loss of generality, assume $e^{f^*} = dP/dQ$. Then we have

$$\text{Var}_Q[e^{f^*}] = \mathbb{E}_Q\left[\left(\frac{dP}{dQ}\right)^2\right] - \left(\mathbb{E}_Q\left[\frac{dP}{dQ}\right]\right)^2 = \mathbb{E}_P\left[\frac{dP}{dQ}\right] - 1, \quad (8)$$

since $\mathbb{E}_Q[dP/dQ] = 1$. Then the variance of $n$ i.i.d random variable gives us

$$\text{Var}_Q\left[\mathbb{E}_{Q_n}[e^{f^*}]\right] = \frac{\text{Var}[e^{f^*}]}{n} \to 0 \text{ as } n \to \infty$$

For a sequence of random variable $X_n$ defined with respect to distribution $Q$, and assume we have $\lim_{n\to\infty} X_n = \mathbb{E}[X]$, then the following comes from the delta method:

$$\lim_{n\to\infty} n \cdot \text{Var}_Q[f(X_n)] = (f'(\mathbb{E}[X]))^2 \cdot \text{Var}_Q[X]. \quad (9)$$

Thus, by applying $f(t) = t^\gamma$ and $\mathbb{E}_Q[dP/dQ] = 1$ gives us

$$\begin{aligned}
\lim_{n\to\infty} n \cdot \text{Var}_Q[\mathbb{E}_{Q_n}[e^{\gamma f^*}]] &= \gamma^2 \cdot \text{Var}_Q[e^{f^*}] \\
&= \gamma^2(\mathbb{E}_P[dP/dQ] - 1) \\
&\geq e^{\gamma^2 \mathbb{E}_P[\log(dP/dQ)]} - \gamma^2 \\
&= e^{\gamma^2 D_{\text{KL}}(P\|Q)} - \gamma^2,
\end{aligned} \quad (10)$$

where the last two equations are from Jensen's inequality and the definition of KL divergence. Now, by applying $f(x) = \log x$ in (9), we have

$$\begin{aligned}
\lim_{n\to\infty} n \cdot \text{Var}_Q[\log \mathbb{E}_{Q_n}[e^{\gamma f^*}]] &= \lim_{n\to\infty} \frac{n \cdot \text{Var}_Q[\mathbb{E}_{Q_n}[e^{\gamma f^*}]]}{(\mathbb{E}_{Q_n}[e^{\gamma f^*}])^2} \\
&= \lim_{n\to\infty} \frac{n \cdot \text{Var}_Q[\mathbb{E}_{Q_n}[e^{\gamma f^*}]]}{e^{2\gamma(\gamma-1)R_\gamma(P\|Q)}} \\
&\geq \frac{e^{\gamma^2 D_{\text{KL}}(P\|Q)} - \gamma^2}{e^{2\gamma(\gamma-1)R_\gamma(P\|Q)}}
\end{aligned}$$

from the definition of Rényi divergence and (10). Thus, we have the following:

$$\lim_{n\to\infty} n \cdot \text{Var}[\mathcal{I}_{\text{Renyi}}^{m,n}[e^{f^*}]] \geq \lim_{n\to\infty} n \cdot \text{Var}[\log \mathbb{E}_{Q_n}[e^{f^*}]] \geq \frac{e^{\gamma^2 D_{\text{KL}}(P\|Q)} - \gamma^2}{e^{2\gamma(\gamma-1)R_\gamma(P\|Q)}}.$$

## A.2 Proof of Theorem 4.2

To prove Theorem 4.2, we first state following Lemma.

**Lemma A.1.** *(Proposition 1 & 2 in [24]) Given positive integers $n \geq 1$ and $m \geq 2$, and for any collection of positive random variables $\{X_i\}_{i=1}^n$ and $\{Y_{i,j}\}_{j=1}^K$ for each $i = 1, \ldots, n$ such that $X_i, Y_{i,1}, \ldots, Y_{i,K}$ are exchangeable. Then for any $\alpha \in (0, \frac{2}{K+1}]$, the following inequality holds:*

$$\mathbb{E}\left[\frac{1}{n}\sum_{i=1}^n \frac{X_i}{\alpha X_i + \frac{1-\alpha}{K}\sum_{i=1}^K Y_{i,j}}\right] \leq \frac{1}{\alpha(K+1)}$$

*Also, for any $\alpha \in \left[\frac{1}{K+1}, \frac{1}{2}\right]$, the following inequality holds:*

$$\mathbb{E}\left[\frac{1}{n}\sum_{i=1}^n \frac{X_i}{\alpha X_i + \frac{1-\alpha}{K}\sum_{i=1}^K Y_{i,j}}\right] \leq 1$$

***Proof of Theorem 4.2.*** First, remark that the DV bound of $\alpha$-skew KL divergence admits following:

$$D_{\text{KL}}^{(\alpha)}(P \,\|\, Q) = \sup_{f \in \mathcal{F}} \ \mathbb{E}_P[f] - \log\left(\alpha \mathbb{E}_P[e^f] + (1-\alpha)\mathbb{E}_Q[e^f]\right). \qquad (11)$$

Then, note that $I_{\text{CPC}}^{(\alpha)}(f)$ satisfies following:

$$
\begin{aligned}
I_{\text{CPC}}^{(\alpha)}(f) &= \mathbb{E}_{(x,y) \sim P_{X,Y},\, y_i \sim P_Y,\, i=1,\ldots,K}\left[\log \frac{e^{f(x,y)}}{\alpha e^{f(x,y)} + \frac{1-\alpha}{K}\sum_{i=1}^{K} e^{f(x,y_i)}}\right] \\
&= \mathbb{E}_{(x,y) \sim P_{X,Y}}[f(x,y)] - \mathbb{E}_{(x,y) \sim P_{X,Y},\, y_i \sim P_Y,\, i=1,\ldots,K}\left[\log\left(\alpha e^{f(x,y)} + \frac{1-\alpha}{K}\sum_{i=1}^{K} e^{f(x,y_i)}\right)\right] \\
&\geq \mathbb{E}_{P_{X,Y}}[f(x,y)] - \log\left(\alpha \mathbb{E}_{P_{X,Y}}[e^{f(x,y)}] + (1-\alpha)\mathbb{E}_{P_X P_Y}[e^{f(x,y)}]\right),
\end{aligned}
$$
$$(12)$$

where the last inequality comes from the Jensen's inequality that $-\mathbb{E}[\log X] \geq -\log \mathbb{E}[X]$. Then since (12) holds for all $f \in \mathcal{F}$, we have

$$\sup_{f \in \mathcal{F}} I_{\text{CPC}}^{(\alpha)}(f) \geq \sup_{f \in \mathcal{F}} \mathbb{E}_{P_{X,Y}}[f(x,y)] - \log\left(\alpha \mathbb{E}_{P_{X,Y}}[e^{f(x,y)}] + (1-\alpha)\mathbb{E}_{P_X P_Y}[e^{f(x,y)}]\right) = D_{\text{KL}}^{(\alpha)}\left(P_{X,Y} \,\|\, P_X P_Y\right).$$

Also, KL divergence also admits following variational form as known as NWJ objective [79]:

$$D_{\text{KL}}(P \,\|\, Q) = \sup_{g \in \mathcal{F}} \ \mathbb{E}_P[g] + 1 - \mathbb{E}_Q[e^g].$$

Then by taking $g(x,y) = \log \frac{e^{f(x,y)}}{\alpha e^{f(x,y)} + \frac{1-\alpha}{K}\sum_{i=1}^{K} e^{f(x,y_i)}}$ for some sampled negatives $y_i \sim P_Y, i = 1,\ldots,K$, in NWJ objective, we have following:

$$
\begin{aligned}
D_{\text{KL}}^{(\alpha)}(P_{X,Y} \,\|\, P_X P_Y) &\geq \mathbb{E}_{y_i \sim P_Y,\, i=1,\ldots,K}\left[\mathbb{E}_{P_{X,Y}}\left[\log \frac{e^{f(x,y)}}{\alpha e^{f(x,y)} + \frac{1-\alpha}{K}\sum_{i=1}^{K} e^{f(x,y_i)}}\right] + 1 \right. \\
&\quad - \alpha \mathbb{E}_{P_{X,Y}}\left[\frac{e^{f(x,y)}}{\alpha e^{f(x,y)} + \frac{1-\alpha}{K}\sum_{i=1}^{K} e^{f(x,y_i)}}\right] \\
&\quad \left. - (1-\alpha)\mathbb{E}_{P_X P_Y}\left[\frac{e^{f(x,y)}}{\alpha e^{f(x,y)} + \frac{1-\alpha}{K}\sum_{i=1}^{K} e^{f(x,y_i)}}\right]\right] \\
&\geq \mathbb{E}_{(x,y) \sim P_{X,Y},\, y_i \sim P_Y,\, i=1,\ldots,K}\left[\log \frac{e^{f(x,y)}}{\alpha e^{f(x,y)} + \frac{1-\alpha}{K}\sum_{i=1}^{K} e^{f(x,y_i)}}\right] \\
&= I_{\text{CPC}}^{(\alpha)}(f),
\end{aligned}
$$

by Lemma A.1, we have

$$1 - \alpha \mathbb{E}_{P_{X,Y}}\left[\frac{e^{f(x,y)}}{\alpha e^{f(x,y)} + \frac{1-\alpha}{K}\sum_{i=1}^{K} e^{f(x,y_i)}}\right] - (1-\alpha)\mathbb{E}_{P_X P_Y}\left[\frac{e^{f(x,y)}}{\alpha e^{f(x,y)} + \frac{1-\alpha}{K}\sum_{i=1}^{K} e^{f(x,y_i)}}\right] \geq 1 - \frac{1}{\alpha(K+1)} \geq 0,$$

for $\alpha \in \left(0, \frac{2}{K+1}\right)$, and

$$1 - \alpha \mathbb{E}_{P_{X,Y}}\left[\frac{e^{f(x,y)}}{\alpha e^{f(x,y)} + \frac{1-\alpha}{K}\sum_{i=1}^{K} e^{f(x,y_i)}}\right] - (1-\alpha)\mathbb{E}_{P_X P_Y}\left[\frac{e^{f(x,y)}}{\alpha e^{f(x,y)} + \frac{1-\alpha}{K}\sum_{i=1}^{K} e^{f(x,y_i)}}\right] \geq 1 - 1 \geq 0,$$

for $\alpha \in \left[\frac{1}{K+1}, \frac{1}{2}\right]$. Therefore, since we have

$$D_{\text{KL}}^{(\alpha)}(P \,\|\, Q) \leq \sup_{f \in \mathcal{F}} I_{\text{CPC}}^{(\alpha)}(f) \leq D_{\text{KL}}^{(\alpha)}(P \,\|\, Q),$$

we have our results.

## A.3 Proof of Theorem 4.3

In this section, we state and define the regularity assumptions to derive the asymptotic upper bounds for the variance of $\alpha$-skew KL divergence in Theorem 3.2. We first review the definition of $f$-divergence. Let $f : (0, \infty) \to \mathbb{R}$ be a convex function with $f(1) = 0$. Let $P$ and $Q$ be distributions with respect to a base measure $dx$ on domain $\mathcal{X}$, and assume $P \ll Q$. Then the $f$-divergence generated by $f$ is defined by

$$D_f(P\|Q) := \mathbb{E}_Q\left[f\left(\frac{dP}{dQ}\right)\right].$$

Note that for any $c \in \mathbb{R}$, $D_{f_c}(P\|Q) = D_f(P\|Q)$, where $f_c(t) = f(t) + c(t-1)$. Hence, w.l.o.g, we assume $f(t) \geq 0$ for all $t \in (0, \infty)$. The conjugate of $f$ is a function $f^* : (0, \infty) \to [0, \infty)$ defined by $f^*(t) = tf(1/t)$, where $f^*(0) = \lim_{t\to 0^+} f^*(t)$ for convenience. The conjugate $f^*$ of convex function $f$ is also convex. Also, one can see that $f^*(1) = 0$ and $f^*(t) \geq 0$ for all $t \in (0, \infty)$, thus it induces another divergence $D_{f^*}$. The conjugate divergence $D_{f^*}$ satisfies $D_{f^*}(P\|Q) = D_f(Q\|P)$.

The KL divergence is a $f$-divergence generated by $f(t) = t\log t - t + 1$, and the $\alpha$-skew KL divergence is a $f$-divergence generated by

$$f^{(\alpha)}(t) = t\log\left(\frac{t}{\alpha t + 1 - \alpha}\right) - (1 - \alpha)(t - 1).$$

From [80], we state following regularity assumptions on the functions $f$ and $f^*$.

**Assumption A.1.** *The generator $f$ is twice continuously differentiable with $f'(1) = 0$. Moreover*

*(A1) We have $C_0 := f(0) < \infty$ and $C_0^* := f^*(0) < \infty$.*

*(A2) There exist constants $C_1, C_1^* < \infty$ such that for any $t \in (0, 1)$, we have,*

$$|f'(t)| \leq C_1 \max\{1, \log(1/t)\}, \quad \text{and} \quad |(f^*)'(t)| \leq C_1^* \max\{1, \log(1/t)\}.$$

*(A3) There exist constants $C_2, C_2^* < \infty$ such that for every $t \in (0, \infty)$, we have,*

$$\frac{t}{2}f''(t) \leq C_2, \quad \text{and} \quad \frac{t}{2}(f^*)''(t) \leq C_2^*.$$

We refer authors [80] for the detailed discussion on Assumption A.1. Then one can observe that KL divergence does not satisfy Assumption A.1, because KL divergence can be unbounded. On the other hand, the $\alpha$-skew KL divergence satisfies Assumption A.1 from following proposition.

**Proposition A.1** ([80]). *The $\alpha$-skew KL divergence generated by $f^{(\alpha)}$ satisfies Assumption A.1 with*

$$C_0 = 1 - \alpha, \quad C_0^* = \log\frac{1}{\alpha} - 1 + \alpha, \quad C_1 = 1, \quad C_1^* = \frac{(1-\alpha)^2}{\alpha}, \quad C_2 = \frac{1}{2}, \quad C_2^* = \frac{1-\alpha}{8\alpha}.$$

For the general $f$-divergences which satisfy Assumption A.1, the following concentration bound holds.

**Proposition A.2** ([80]). *Assume $f$ satisfies Assumption A.1, and let $P$ and $Q$ be two distributions with $P \ll Q$. Let $P_m$ be $m$ i.i.d samples from $P$ and $Q_n$ be $n$ i.i.d samples from $Q$. Then the $f$-divergence $D_f$ satisfies following:*

$$\mathbb{P}[\,|D_f(P_m\|Q_n) - \mathbb{E}[D_f(P_m\|Q_n)]| > \varepsilon\,] \leq 2\exp\left(-\frac{\varepsilon^2}{\frac{2}{m}(C_1 \log m + c_1)^2 + \frac{2}{n}(C_1^* \log n + c_2)^2}\right)$$

*where $c_1 = \max\{C_0^*, C_2\}$ and $c_2 = \max\{C_0, C_2^*\}$.*

Thus, the following lemma derives a concentration bound for the $\alpha$-skew KL divergence by plugging the constants in Proposition A.1 to Proposition A.2.

**Lemma A.2.** *For $\alpha < \frac{1}{8}$, the following holds:*

$$\mathbb{P}[\,|D_{KL}^{(\alpha)}(P_m\|Q_n) - \mathbb{E}[D_{KL}^{(\alpha)}(P_m\|Q_n)]| > \varepsilon\,] \leq 2\exp\left(-\frac{\varepsilon^2}{\frac{2}{m}\log^2(\alpha m) + \frac{2}{\alpha^2 n}\log^2(e^{1/8}n)}\right)$$

*Proof.* Note that $C_0^* = \log(1/\alpha) - 1 + \alpha \geq C_2 = 1/2$, and $C_0 = 1 - \alpha \leq C_2^* = \frac{1-\alpha}{8\alpha}$ for $\alpha < \frac{1}{8}$. Then the concentration bound follows from Proposition A.2. □

Lastly, we present following upper bound on the bias of empirical estimator of KL divergence:

**Proposition A.3** ([81]). *Suppose $P \ll Q$, and $\mathrm{Var}[dP/dQ] < \infty$. Then we have*

$$|\mathbb{E}[D_{KL}(P_m\|Q_n)] - D_{KL}(P\|Q)| \leq \frac{\chi^2(P\|Q)}{\min\{n, m\}}.$$

From proposition A.3, we have

$$|\mathbb{E}[D_{\mathrm{KL}}^{(\alpha)}(P_m\|Q_n)] - D_{\mathrm{KL}}^{(\alpha)}(P\|Q)| \leq \frac{\chi^2(P\|\alpha P + (1-\alpha)Q)}{\min\{n, m\}},$$

where $\chi^2(P\|\alpha P + (1-\alpha)Q) = \int \frac{d^2 P}{\alpha dP + (1-\alpha)dQ} \leq \int \frac{1}{\alpha} dP = \frac{1}{\alpha}$, or $\int \frac{d^2 P}{\alpha dP + (1-\alpha)dQ} \leq \frac{1}{1-\alpha} \int \frac{d^2 P}{dQ} = \frac{\chi^2(P\|Q)}{1-\alpha}$. Therefore, we have

$$|\mathbb{E}[D_{\mathrm{KL}}^{(\alpha)}(P_m\|Q_n)] - D_{\mathrm{KL}}^{(\alpha)}(P\|Q)| \leq \frac{c(\alpha)}{\min\{n, m\}}, \quad \text{for} \quad c(\alpha) := \min\left\{\frac{1}{\alpha}, \frac{\chi^2(P\|Q)}{1-\alpha}\right\}.$$

Now we present the proof of Theorem 4.3 in the main paper.

*Proof of Theorem 4.3.* Define

$$B_1 := |D_{\mathrm{KL}}^{(\alpha)}(P_m\|Q_n) - \mathbb{E}[D_{\mathrm{KL}}^{(\alpha)}(P_m\|Q_n)]|$$
$$B_2 := |\mathbb{E}[D_{\mathrm{KL}}^{(\alpha)}(P_m\|Q_n)] - D_{\mathrm{KL}}^{(\alpha)}(P\|Q)|$$
$$B_3 := |\mathcal{I}_{\mathrm{KL}}^{(\alpha)}(\widehat{f}) - D_{\mathrm{KL}}^{(\alpha)}(P_m\|Q_n)|.$$

From Proposition A.3, we have $B_2 \leq \frac{c_2}{\min\{n, m\}}$ for some constant $c(\alpha) > 0$, and from the assumption, we have $B_3 \leq \varepsilon_f$. By using triangle inequality twice, we have

$$B_1 \geq |D_{\mathrm{KL}}^{(\alpha)}(P_m\|Q_n) - D_{\mathrm{KL}}^{(\alpha)}(P\|Q)| - B_2$$
$$\geq |D_{\mathrm{KL}}^{(\alpha)}(P\|Q) - \mathcal{I}_{\mathrm{KL}}^{(\alpha)}(\widehat{f})| - B_3 - B_2.$$

Therefore, it follows that

$$\mathbb{P}[|D_{\mathrm{KL}}^{(\alpha)}(P\|Q) - \mathcal{I}_{\mathrm{KL}}^{(\alpha)}(\widehat{f})| > \varepsilon] \leq \mathbb{P}[B_1 + B_2 + B_3 > \varepsilon]$$
$$\leq \mathbb{P}\left[B_1 > \varepsilon - \varepsilon_f - \frac{c(\alpha)}{\min\{n, m\}}\right]$$
$$\leq 2\exp\left(-\frac{(\varepsilon - \varepsilon_f - c(\alpha)/\min\{n, m\})^2}{\frac{2}{m}\log^2(\alpha m) + \frac{2}{\alpha^2 n}\log^2(e^{1/8}n)}\right),$$

where the last inequality comes from Proposition A.2. Since the following holds for any random variable $X$,

$$\mathrm{Var}(X) = \mathbb{E}[(X - \mathbb{E}X)^2] = \int_0^\infty \mathbb{P}[|X - \mathbb{E}X|^2 > t]dt = \int_0^\infty \mathbb{P}[|X - \mathbb{E}X| > \sqrt{t}]dt,$$

we have

$$\mathrm{Var}_{P,Q}\left[\mathcal{I}_{\mathrm{KL}}^{(\alpha)}(\widehat{f})\right] \leq \int_0^\infty 2\exp\left(-\frac{(\sqrt{t} - \varepsilon_f - c(\alpha)/\min\{n, m\})^2}{\frac{2}{m}\log^2(\alpha m) + \frac{2}{\alpha^2 n}\log^2(e^{1/8}n)}\right) dt,$$

which proves our result.

## A.4 Proof sketch for Rényi divergence

In this section, we provide a proof sketch for the upper bound for the variance of RMLCPC objective. We use similar proof technique that we used in previous section. Recall that the Rényi Divergence is related to $\gamma$-divergence[3] $D_\gamma$, a $f$-divergence generated by $f^{(\gamma)}(t) = \frac{t^\gamma - 1}{\gamma(\gamma-1)}$. The KL divergence is recovered from $\gamma$-divergence if we let $\gamma \to 1$. Then we have following equation for Rényi divergence and $\gamma$-divergence:

$$R_\gamma(P\|Q) = \frac{1}{\gamma(\gamma-1)} \log(\gamma(\gamma-1)D_\gamma(P\|Q) + 1).$$

One can observe that $f(x) = \frac{1}{\gamma(\gamma-1)} \log(\gamma(\gamma-1)x + 1)$ is of 1-Lipschitz function for any $\gamma \neq 0, 1$. Thus, if we can derive concentration bound for $\alpha$-skew $\gamma$-divergence, we can derive concentration bound for Rényi divergence.

Here, we provide an example when $\gamma = 2$. Remark that when $\gamma = 2$, the $\gamma$-divergence is equivalent to $\chi^2$-divergence, which is generated by $f(t) = (t-1)^2$. Then the $\alpha$-skew $\chi^2$-divergence is generated by following generators:

$$f^{(\alpha)}(t) = \frac{(t-1)^2}{\alpha t + 1 - \alpha} = (f^{(1-\alpha)})^*(t).$$

Then from [80], the $\alpha$-skew $\chi^2$-divergence satisfies Assumption A.1 with

$$C_0 = \frac{1}{1-\alpha}, \quad C_0^* = \frac{1}{\alpha}, \quad C_1 = \frac{2}{(1-\alpha^2)}, \quad C_1^* = \frac{2}{\alpha^2}, \quad C_2 = \frac{4}{27\alpha(1-\alpha)^2}, \quad C_2^* = \frac{4}{27\alpha^2(1-\alpha)}.$$

Then by using Proposition A.2, for $\alpha < \min\{\frac{2}{3\sqrt{3}}, 1 - \frac{2}{3\sqrt{3}}\}$, we have

$$\mathbb{P}[|D_{\chi^2}^{(\alpha)}(P_m\|Q_n) - \mathbb{E}[D_{\chi^2}^{(\alpha)}(P_m\|Q_n)]| > \varepsilon] \leq 2\exp\left(-\frac{\varepsilon^2}{\frac{c_1}{n}\log^2(c_2 n) + \frac{c_3}{\alpha^2 m}\log^2(c_4 m)}\right),$$

for some constant $c_1, c_2, c_3, c_4 > 0$, then from the Lipschitz continuity, we have

$$\mathbb{P}[|R_2^{(\alpha)}(P_m\|Q_n) - \mathbb{E}[R_2^{(\alpha)}(P_m\|Q_n)]| > \varepsilon] \leq 2\exp\left(-\frac{\varepsilon^2}{\frac{c_1}{n}\log^2(c_2 n) + \frac{c_3}{\alpha^2 m}\log^2(c_4 m)}\right),$$

Thus, by deriving the asymptotic bound for the bias $|\mathbb{E}[R_2^{(\alpha)}(P_m\|Q_n)] - R_2^{(\alpha)}(P\|Q)|$, and if the Rényi variational objective has sufficiently small error, we can derive upper bound for the variance of RMLCPC objective.

# B Details for Experiments

## B.1 Implementation

As explained in Section 3.3, we derive the full equivalent form of MLCPC and RMLCPC with non-zero $\alpha$. Recall that the $\alpha$-MLCPC objective with neural network $f_\theta$ is given by

$$\mathcal{I}_{\text{MLCPC}}^{(\alpha)}(f_\theta) = \mathbb{E}_{v,v^+}[f_\theta(v, v^+)] - \log\left(\alpha\mathbb{E}_{v,v^+}[e^{f_\theta(v,v^+)}] + (1-\alpha)\mathbb{E}_{v,v^-}[e^{f_\theta(v,v^-)}]\right), \quad (13)$$

then the gradient of (13) with respect to parameter $\theta$ is given by

$$\nabla_\theta \mathcal{I}_{\text{MLCPC}}^{(\alpha)}(f_\theta)$$

$$= \mathbb{E}_{v,v^+}[\nabla_\theta f_\theta(v, v^+)] - \frac{\alpha\mathbb{E}_{v,v^+}[e^{f_\theta(v,v^+)}\nabla_\theta f_\theta(v, v^+)] + (1-\alpha)\mathbb{E}_{v,v^-}[e^{f_\theta(v,v^-)}\nabla_\theta f_\theta(v, v^-)]}{\alpha\mathbb{E}_{v,v^+}[e^{f_\theta(v,v^+)}] + (1-\alpha)\mathbb{E}_{v,v^-}[e^{f_\theta(v,v^-)}]}$$

$$= \mathbb{E}_{v,v^+}[\nabla_\theta f_\theta(v, v^+)] - \left(\mathbb{E}_{\text{sg}(q_\theta(v,v^+))}[\nabla_\theta f_\theta(v, v^+)] + \mathbb{E}_{\text{sg}(q_\theta(v,v^-))}[\nabla_\theta f_\theta(v, v^-)]\right),$$

---

[3]we use $\gamma$ for consistency, in general, it is called $\alpha$-divergence.

Table 7: Ablation on the effect of $\alpha$ when training with harder data augmentation.

| $\alpha^{-1}$ | 1024 | 4096 | 16384 | 65536 |
|---|---|---|---|---|
| Base Aug. | 79.0 | 79.3 | 78.6 | 78.4 |
| Hard Aug. | 81.1 | 81.3 | **81.6** | 81.1 |
| Gap | +2.1 | +2.0 | **+3.0** | +2.7 |

where

$$q_\theta(v, v^+) \propto \frac{\alpha p(v, v^+) e^{f_\theta(v, v^+)}}{\alpha \mathbb{E}_{v,v^+}[e^{f_\theta(v,v^+)}] + (1-\alpha)\mathbb{E}_{v,v^-}[e^{f_\theta(v,v^-)}]}$$

$$q_\theta(v, v^-) \propto \frac{(1-\alpha) p(v) p(v^-) e^{f_\theta(v, v^-)}}{\alpha \mathbb{E}_{v,v^+}[e^{f_\theta(v,v^+)}] + (1-\alpha)\mathbb{E}_{v,v^-}[e^{f_\theta(v,v^-)}]},$$

are importance weights for positive and negative pairs in the second term of (13). For the generalized $(\alpha, \gamma)$-RMLCPC, recall that the RMLCPC with neural network $f_\theta$ is given by

$$\mathcal{I}_{\text{RMLCPC}}^{(\alpha,\gamma)}(f_\theta) = \frac{1}{\gamma-1}\log\mathbb{E}_{v,v^+}[e^{(\gamma-1)f_\theta(v,v^+)}] - \frac{1}{\gamma}\log\left(\alpha\mathbb{E}_{v,v^+}[e^{\gamma f_\theta(v,v^+)}] + (1-\alpha)\mathbb{E}_{v,v^-}[e^{\gamma f_\theta(v,v^-)}]\right). \tag{14}$$

Then the gradient of (14) with respect to $\theta$ is given by

$$\nabla_\theta \mathcal{I}_{\text{RMLCPC}}^{(\alpha,\gamma)}(f_\theta)$$

$$= \mathbb{E}_{q_\theta^{(1)}(v,v^+)}[\nabla_\theta f_\theta(v,v^+)] - \frac{\alpha\mathbb{E}_{v,v^+}[e^{\gamma f_\theta(v,v^+)}\nabla_\theta f_\theta(v,v^+)] + (1-\alpha)\mathbb{E}_{v,v^-}[e^{\gamma f_\theta(v,v^-)}\nabla_\theta f_\theta(v,v^-)]}{\alpha\mathbb{E}_{v,v^+}[e^{\gamma f_\theta(v,v^+)}] + (1-\alpha)\mathbb{E}_{v,v^-}[e^{\gamma f_\theta(v,v^-)}]}$$

$$= \mathbb{E}_{\text{sg}(q_\theta^{(1)}(v,v^+))}[\nabla_\theta f_\theta(v,v^+)] - \left(\mathbb{E}_{\text{sg}(q_\theta^{(2)}(v,v^+))}[\nabla_\theta f_\theta(v,v^+)] + \mathbb{E}_{\text{sg}(q_\theta^{(2)}(v,v^-))}[\nabla_\theta f_\theta(v,v^-)]\right),$$

where

$$q_\theta^{(1)}(v, v^+) \propto p(v, v^+) e^{(\gamma-1)f_\theta(v,v^+)}$$

$$q_\theta^{(2)}(v, v^+) \propto \frac{\alpha p(v, v^+) e^{\gamma f_\theta(v,v^+)}}{\alpha\mathbb{E}_{v,v^+}[e^{\gamma f_\theta(v,v^+)}] + (1-\alpha)\mathbb{E}_{v,v^-}[e^{\gamma f_\theta(v,v^-)}]}$$

$$q_\theta^{(2)}(v, v^-) \propto \frac{(1-\alpha) p(v) p(v^-) e^{\gamma f_\theta(v,v^-)}}{\alpha\mathbb{E}_{v,v^+}[e^{\gamma f_\theta(v,v^+)}] + (1-\alpha)\mathbb{E}_{v,v^-}[e^{\gamma f_\theta(v,v^-)}]},$$

are importance weights for positive and negative pairs for the first and second term in (14). As discussed in Section 3.3, regardless of the insertion of parameter $\alpha$, MLCPC and RMLCPC explicitly conduct hard-negative sampling, and especially RMLCPC, conducts easy-positive sampling.

The PyTorch style pseudo-code for our implementation on RényiCL is demonstrated in Algorithm 1. In our default implementation, we use multi-crop data augmentation [43] with two global views and multiple local views. In Algorithm 1, we implement the negative of $(\alpha, \gamma)$-RMLCPC, and $\alpha$-MLCPC which is equivalent to the case when $\gamma \to 1$ for RMLCPC, for the contrastive losses. Note that our pseudo-code for contrastive objectives are based on the approach that uses importance weights.

**Effect of $\alpha$ in RényiCL.** One can observe that the gradient of positive pairs in the second term is multiplied by $\alpha$, thus the gradient of positive pairs is affected by $\alpha$ and negative pairs. However, since we use a small value of $\alpha$, the gradient of the second term can be ignored in practice. To verify our claim, we experimented with different values of $\alpha$ on both basic and hard data augmentations:

In Table 7, one can observe that as $\alpha$ becomes smaller, the gap between using hard augmentation and the base augmentation becomes larger. Thus, there is a tradeoff in the choice of $\alpha$: the $\alpha$ should be large enough to evade large variance (Thm 3.2), but small $\alpha$ is preferred to have the effect of easy positive sampling.

## B.2 Further ablation studies

In this section, we provide further ablation studies to validate the effectiveness of RényiCL.

**Algorithm 1** RényiCL: PyTorch-like Pseudocode

```
# f_q: base encoder: backbone + proj mlp + pred mlp
# f_k: momentum encoder: backbone + proj mlp
# m: momentum coefficient
# tau: temperature
# aug_g : global view data augmentation
# aug_l : local view data augmentation

for x in loader: # load a minibatch x with N samples
    x1, x2 = aug_g(x), aug_g(x) # two global views
    k1, k2 = f_k(x1), f_k(x2) # keys: [N, D] each
    q1, q2 = [], [] # list of queries
    q1.append(f_q(x1)), q2.append(f_q(x2)) # queries: [N, D] each
    for i in range(n_crops):
        x_l = aug_l(x)
        q1.append(f_q(x_l)), q2.append(f_q(x_l)) # pass local views only to base encoder

    pos1, pos2, neg1, neg2 = [], [], [], []
    for j in range(n_crops + 1):
        pos, neg = extract_pos_neg(q1[j], k2) # extract pos, neg from q1 and k2
        pos1.append(pos)
        neg1.append(neg)

        pos, neg = extract_pos_neg(q2[j], k1) # extract pos, neg from q2 and k1
        pos2.append(pos)
        neg2.append(neg)
    pos1, pos2, neg1, neg2 = pos1.cat(), pos2.cat(), neg1.cat(), neg2.cat()
    loss = ctr_loss(pos1, neg1) + ctr_loss(pos2, neg2) # symmetrized

    update(f_q) # optimizer update: f_q
    f_k = m*f_k + (1-m)*f_q # momentum update: f_k

# extract positive pairs and negative pairs
def extract_pos_and_neg(q, k):
    # N : number of samples in minibatch
    logits = mm(q, k.t()) / tau # matrix multiplication
    pos = logits.diag() # size N
    neg = logits.flatten()[1:].view(N-1, N+1)[:,:-1].reshape(N,N-1) # size N x (N-1)
    return pos, neg

# contrastive losses using importance weights
def ctr_loss(pos, neg, alpha, gamma):
    pos_d = pos.detach(), neg_d = neg.detach() # no gradient for importance weights
    if gamma == 1: # MLCPC
        loss_1 = -1 * pos.mean()
        iw2_p, iw2_n = pos_d.exp(), neg_d.exp() # importance weights
        loss_2 = alpha*(pos*iw2_p.exp()).mean()+(1-alpha)*(neg*iw2_n.exp()).mean()
        loss_2 /= alpha*iw2_p.mean() + (1-alpha)*iw2_n.mean()
        loss = loss_1 + loss_2
    elif gamma > 1: # RMLCPC
        iw1 = ((gamma - 1)*pos_d).exp() # importance weight for first term
        loss_1 = (pos*iw1).mean() / iw1.mean()
        iw2_p, iw2_n = (gamma*pos_d).exp(), (gamma*neg_d).exp() # importance weights
        loss_2 = alpha*(pos*iw2_p).mean() + (1-alpha)*(neg*iw2_n).mean()
        loss_2 /= alpha*iw2_p.mean() + (1-alpha)*iw2_n.mean()
        loss = loss_1 + loss_2
    return loss
```

**Fair comparison with other self-supervised methods.** To see the effectiveness of RényiCL, we conduct additional experiments for fair comparison with other self-supervised methods. MoCo v3 [6] is a state-of-the-art method in contrastive self-supervised learning. For fair comparison of RényiCL with MoCo v3, we apply harder data augmentations such as RandAugment [50] and multi-crops [43] on MoCo v3. We first reimplemented MoCo v3 with smaller batch size (original 4096 to 1024) and in our setting, we achieved 69.6%, which is better than their original reports (68.9%). Then we use harder data augmentations RandAugment (RA), RandomErasing (RE), and multi-crops (MC) when training MoCo v3. Second, we compare RényiCL with DINO [82]. Note that DINO uses default multi-crops, therefore we further applied RandAugment (RA) and RandomErasing (RE) for a fair comparison. For each model, we train for 100 epochs. The results are shown in Table 8. One can observe that while MoCo v3 and DINO benefit from using harder data augmentation, RenyiCL shows the best performance when using harder data augmentation. Also, compared to the Base data augmentation, RényiCL attains the most gain by using harder data augmentation.

**Robustness to data augmentations.** To see the robustness of RényiCL on the choice of data augmentation, we conduct additional ablation studies on the robustness of RényiCL on various data

Table 8: Fair comparison with other self-supervised learning methods by using same hard data augmentation. All models are run by us with 100 training epochs.

| Method | DINO [82] | MoCo v3 [6] | RényiCL |
|--------|-----------|-------------|---------|
| Base | 70.9 | 69.6 | 69.4 |
| Hard | 72.5 | 73.5 | **74.3** |
| Gain | +1.6 | +3.9 | **+4.9** |

Table 9: Ablation on the robustness of RényiCL to Noisy Crop.

| Method | CIFAR-10 | CIFAR-100 | ImageNet-100 |
|--------|----------|-----------|--------------|
| CPC | 91.7 | 65.4 | 79.7 |
| +NC | 90.6(-1.1) | 64.9(-0.5) | 79.2(-0.5) |
| MLCPC | 91.9 | 65.6 | 79.5 |
| +NC | 90.7(-1.2) | 65.0(-0.6) | 79.4(-0.1) |
| RMLCPC | 90.7 | 64.5 | 78.9 |
| +NC | **91.6(+0.9)** | **67.6(+3.1)** | **80.5(+1.6)** |

Table 10: Ablation on the robustness of RényiCL when learning with limited data augmentation.

| Method | CIFAR-10 | CIFAR-100 | STL-10 |
|--------|----------|-----------|--------|
| CPC | 63.2 | 31.8 | 52.8 |
| MLCPC | 63.3 | 32.9 | 53.1 |
| RMLCPC | **67.1** | **41.2** | **56.4** |

augmentation schemes. We first show that RényiCL is more robust to the noisy data augmentation, as similar to that in [30], we apply Noisy Crop, which additionally perform RandomResizedCrop of size 0.2. Then the generated views might contain only nuisance information (e.g. background), thus it can hurt the generalization of contrastive learning. We train each model with CPC, MLCPC, and RMLCPC objectives and data augmentations with and without Noisy Crop. We experimented on CIFAR-10, CIFAR-100, and ImageNet-100 datasets and the results are shown in Table 9. One can observe that while the performance of using CPC and MLCPC degrades as we apply the noisy crop, RMLCPC conversely improves the performance. Hence, RMLCPC is robust in the choice of data augmentation.

Second, we experiment when there is limited data augmentation. For each CIFAR-10, CIFAR-100, and STL-10 dataset, we remove data augmentations such as color jittering and grayscale and only apply RandomResizedCrop and HorizontalFlip. The results are in Table 10. Note that RMLCPC achieves the best performance among CPC and MLCPC, when we do not apply data augmentation. Therefore we show that the RMLCPC objective is not only robust to the hard data augmentation but is also robust when there is limited data augmentation.

### B.3 ImageNet experiments

**Data augmentation.** For the baseline, we follow the good practice in existing works [2, 3, 47, 6], which includes random resized cropping, horizontal flipping, color jittering [2], grayscale conversion [2], Gaussian blurring [3], and solarization [47]. Given the baseline data augmentation, we add the following data augmentation methods:

- RandAugment [50]: a strong data augmentation strategy that includes 14 image-specific transformations such as AutoContrast, Posterize, etc. We use default settings that were used in [50] (2 operations at each iteration and a magnitude scale of 10).

- RandomErasing [65]: a simple data augmentation strategy that randomly masks a box in an image with a random value. The RandomErasing was applied with probability 0.2.

- Multi-crop [43]: a multi-view data augmentation strategy for self-supervised learning, we use two global crops of size 224 and 2 or 6 local crops of size 96. For global crops, we scale the image with $(0.2, 1.0)$ and for the local crops, we scale the image with $(0.05, 0.2)$, which is slightly different from that in [43].

For the implementation of RandAugment and RandomErasing, we use the implementation in [83]. Note that when using Multi-crop, we only use RandAugment on the global views, and we do not use

RandomErasing, which slightly degrades performance when used for Multi-crops. Otherwise, if we do not use Multi-crop, we use both RandAugment and RandomErasing.

**Model.**    We use ResNet-50 [49] for backbone. For our base encoder, we further use a projection head [3], and a prediction head [47]. The momentum encoder is updated by the moving average of the base encoder, except that we do not use the prediction head for momentum encoder [6]. The momentum coefficient is initialized by 0.99 and gradually increased by the half-cycle cosine schedule to 1. We use 2-layer MLP with dimensions 2048-4096-256 for the projection head. We attach the batch-normalization (BN) layer and ReLU activation layer at the end of each fully-connected layer, except for the last one, to which we only attach the BN layer. For the prediction head, we use 2-layer MLP with dimensions of 256-4096-256, and we attach BN and ReLU at the end of each fully connected layer except the last one. Then the critic is implemented by the temperature-scaled cosine similarity: $f(x, y; \tau) = x^\top y / (\tau \|x\|_2 \|y\|_2)$, with $\tau = 0.5$. For RMLCPC objective, we use $\gamma = 2.0$, and $\alpha = 1/65536$.

**Optimization.**    We use LARS [84] optimizer with a batch size of 512, weight decay of $1.5 \times 10^{-6}$, and momentum of 0.9 for all experiments. We use root scaling for learning rate [3], which was effective in our framework. When training for 100 epochs, we use the base learning rate of 0.15, and for training for 200 epochs, we use the base learning rate of 0.075. The learning rate is gradually annealed by a half-cycle cosine schedule. We also used automatic mixed precision for scalability.

**Linear evaluation.**    For ImageNet linear evaluation accuracy, we report the validation accuracy of the linear classifier trained on the top of frozen features. We use random resized cropping of size 224 and random horizontal flipping for training data augmentation, and resizing to the size of 256 and center-crop of 224 for validation data augmentation. We use cross-entropy loss for training, where we train for 90 epochs with SGD optimizer, batch size of 1024, and base learning rate of 0.1, where we use linear learning rate scaling and learning rate follows cosine annealing schedule. We do not use weight decay.

**Semi-supervised learning.**    For semi-supervised learning on ImageNet, we follow the protocol as in [3, 47, 43]. We add a linear classifier on the top of frozen representation, then train both backbone and classifier using either 1% or 10% of the ImageNet training data. For a fair comparison, we use the splits proposed in [3]. We use the same data augmentation and loss function as in linear evaluation. As same as previous works, we use different learning rates for each backbone and classification head. For the 1% setting, we use a base learning rate of 0.01 for backbone and 0.2 for classification head. For the 10% setting, we use a base learning rate of 0.08 for the backbone and 0.03 for the classification head. The learning rate is decayed with a factor of 0.2 at 16 and 20 epochs. We do not use weight decay or other regularization technique. We train for 20 epochs with SGD optimizer, batch size of 512 for both 1% and 10% settings, and do not use weight decay.

**Transfer learning.**    For transfer learning, we train a classifier on the top of frozen representations as done in many previous works [3, 47]. In Table 11, we list the information of the datasets we used; the number of classes, number of samples for each train/val/test split, and evaluation metric. We use the same data augmentation that we used for linear evaluation on the ImageNet dataset. For optimization, we $\ell_2$-regularized L-BFGS, where the regularization parameter is selected from a range of 45 logarithmically spaced values from $10^{-6}$ to $10^5$ using the validation split. After the best hyperparameter is selected, we train the linear classifier using both training and validation splits and report the test accuracy using the metric instructed in Table 3. The maximum number of iterations in L-BFGS is 5000 and we use the previous solution as an initial point, i.e., a warm start, for the next step. For few-shot learning experiments, we perform logistic regression on the top of frozen representations and use $N \times K$ support samples without fine-tuning and data augmentation in a $N$-way $K$-shot episode.

**Extended version of Table 4.**    We provide additional experimental results in the comparison with other contrastive representation learning methods that use stronger augmentations. The InfoMin [14] used RandAugment [50] and Jigsaw cropping for data augmentation and trained for 800 epochs. The CLSA [59] used both RandAugment [50] and pyramidal multi-crop data augmentation, which generates additional crops of size $192 \times 192, 160 \times 160, 128 \times 128$, and $96 \times 96$. For comparison,

Table 11: Dataset information for the transfer learning tasks. For FC100, CUB200, Plant Disease, we perform few-shot learning, otherwise, we perform linear evaluation.

| Dataset | # of classes | Training | Validation | Test | Metric |
|---|---|---|---|---|---|
| CIFAR10 [51] | 10 | 45000 | 5000 | 10000 | Top-1 accuracy |
| CIFAR100 [51] | 100 | 45000 | 5000 | 10000 | Top-1 accuracy |
| Food [52] | 101 | 68175 | 7575 | 25250 | Top-1 accuracy |
| MIT67 [85] | 67 | 4690 | 670 | 1340 | Top-1 accuracy |
| Pets [60] | 37 | 2940 | 740 | 3669 | Mean per-class accuracy |
| Flowers [53] | 102 | 1020 | 1020 | 6149 | Mean per-class accuracy |
| Caltech101 [61] | 101 | 2525 | 505 | 5647 | Mean Per-class accuracy |
| Cars [54] | 196 | 6494 | 1650 | 8041 | Top-1 accuracy |
| Aircraft [55] | 100 | 3334 | 3333 | 3333 | Mean Per-class accuracy |
| DTD (split 1) [56] | 47 | 1880 | 1880 | 1880 | Top-1 accuracy |
| SUN397 (split 1) [57] | 397 | 15880 | 3970 | 19850 | Top-1 accuracy |
| VOC2007 [58] | 20 | 4952 | 2501 | 2510 | Mean average precision |
| FC100 [62] | 20 | - | - | 12000 | Average accuracy |
| CUB200 [50] | 200 | - | - | 11780 | Average accuracy |
| Plant Disease [63] | 38 | - | - | 54305 | Average accuracy |

Table 12: Comparison of contrastive self-supervised methods with harder augmentations by downstream few-shot classification accuracy (%) over 2000 episodes on FC100, CUB200, and Plant disease datasets. $(N, K)$ denotes $N$-way $K$-shot classification. † denotes the usage of multi-crop data augmentation. **Bold** entries denote the best performance.

| Method | FC100 | | CUB200 | | Plant Disease | |
|---|---|---|---|---|---|---|
| | (5,1) | (5,5) | (5,1) | (5,5) | (5,1) | (5,5) |
| InfoMin [14] | 31.80 | 45.09 | 53.81 | 72.20 | 66.11 | 84.12 |
| CLSA [59] | 34.46 | 49.16 | 50.83 | 67.93 | 67.39 | 86.15 |
| CLSA† [59] | 41.09 | 58.38 | 52.87 | 70.92 | 71.57 | 88.94 |
| RényiCL | 36.31 | 53.39 | **59.73** | 82.12 | 80.68 | 94.29 |
| RényiCL† | **42.10** | **60.80** | 58.25 | **82.38** | **83.40** | **95.71** |

we use the checkpoints from their official github repositories[4][5]. In Table 12, we report the few-shot classification accuracies (%) on FC100 [62], CUB200 [50], and Plant Disease [63] datasets, which is the extended version of Table 3 in the main paper. In Table 13, we report the linear evaluation accuracies (%) on various object classification datasets extending the results in Table 3 in the main paper.

## B.4 CIFAR experiments

For CIFAR experiments, we use random resized cropping with a size of $32 \times 32$, random horizontal flipping, color jittering, and random grayscale conversion with probability 0.2 for the base data augmentation. For harder data augmentation, we apply RandAugment [50], where we use 3 operations at each iteration with a magnitude scale of 5, and RandomErasing [65]. We use modified ResNet-18 [49], where the kernel size of the first convolutional layer is converted from $7 \times 7$ to $3 \times 3$, and we do not use Max pooling at the penultimate layer [3]. We attach a 2-layer projection head of size 512-2048-128 with ReLU and BN layer attached at each FC layer, except for the last one. We do not use a prediction head, and the temperature is set to be 0.5 for every CIFAR experiment. We use $\gamma = 1.5$ and $\alpha = 1/4096$ for both CIFAR-10 and CIFAR-100 experiments. We train for 500 epochs with the SGD optimizer, learning rate of 0.5 (without linear scaling), weight decay of 5e-4, and momentum of 0.9. For evaluation, we train a linear classifier on the top of frozen features with 100 epochs with an SGD optimizer, learning rate of 0.3, without using weight decay. We only used random resized crop with a size of 32, and random horizontal flipping for both training and evaluation.

---

[4]InfoMin: https://github.com/HobbitLong/PyContrast
[5]CLSA: https://github.com/maple-research-lab/CLSA

Table 13: Comparison of contrastive self-supervised methods with harder augmentations by downstream linear evaluation accuracy on various datasets. IN denotes ImageNet validation accuracy (%), and † denotes the use of multi-crop data augmentation. **Bold** entries denote the best performance.

| Method | IN | CIFAR10 | CIFAR100 | Food101 | Pets | MIT67 | Flowers | Caltech101 | Cars | Aircraft | DTD | SUN397 |
|--------|-----|---------|----------|---------|------|-------|---------|-----------|------|----------|-----|--------|
| InfoMin [14] | 73.0 | 92.8 | 75.8 | 73.8 | 86.4 | 76.9 | 91.2 | 88.5 | 49.6 | 50.5 | 75.0 | 61.2 |
| CLSA [59] | 72.2 | 94.2 | 77.4 | 73.2 | 85.2 | 77.1 | 90.8 | 91.2 | 48.7 | 51.6 | 75.0 | 62.0 |
| CLSA† [59] | 73.3 | 93.9 | 76.9 | 73.4 | 86.4 | 77.7 | 91.2 | 90.2 | 47.4 | 50.0 | 75.5 | 62.9 |
| RényiCL | 72.6 | 93.8 | 78.8 | 72.7 | 88.4 | 75.8 | 94.2 | **94.0** | 62.1 | 58.8 | 74.6 | 62.1 |
| RényiCL† | **75.3** | **94.4** | **78.9** | **77.5** | **89.2** | **81.1** | **96.2** | **94.0** | **66.4** | **61.1** | **76.3** | **65.9** |

Table 14: Full experimental results for CovType and Higgs-100K datasets. We report top-1 accuracy (%) with linear evaluation. Average over 5 runs. **Bold** entries denote the best performance.

| | CovType | | | | Higgs-100K | | | |
|--------|---------|----|----|-------|------------|----|----|-------|
| Method | No Aug. | RM | FC | RM+FC | No Aug. | RM | FC | RM+FC |
| CPC | 71.6± 0.36 | 69.8± 0.26 | 74.3± 0.44 | 73.6± 0.16 | 64.7± 0.22 | 71.3± 0.08 | 64.9± 0.11 | 71.4± 0.15 |
| MLCPC | 71.7± 0.26 | 70.2± 0.15 | 74.1± 0.39 | 74.0± 0.45 | 64.9± 0.11 | 71.5± 0.06 | 65.3± 0.26 | 71.5± 0.06 |
| RMLCPC ($\gamma$=1.1) | 72.1± 0.31 | 70.9± 0.42 | **74.9±0.28** | 73.8± 0.47 | 65.1± 0.29 | 71.8± 0.17 | 65.2± 0.13 | 71.9± 0.06 |
| RMLCPC ($\gamma$=1.2) | 71.9± 0.58 | 70.8± 0.25 | 74.5± 0.38 | 73.7± 0.37 | 64.5± 0.41 | **72.4±0.13** | 65.3± 0.44 | 72.3± 0.10 |

## B.5 Tabular experiments

For the data augmentation of tabular experiments, we consider simple random masking (RM) noise that was used in [68], and random feature corruption (FC) proposed in [69]. The random masking noise is analogous to RandomErasing [65] that we used in image contrastive learning. The FC data augmentation randomly selects a subset of attributes on each tabular data and mixes it with the attributes from another data. Note that the random corruption is performed in the whole dataset scale [69]. We refer to [69] for detailed implementation of FC data augmentation. We use a probability of 0.2 for both random masking noise and FC data augmentation. For the backbone, we use a 5-layer MLP with a dimension of 2048-2048-4096-4096-8192, where all layers have a BN layer followed by RELU, except for the last one. We use MaxOut activation for the last layer with 4 sets, and on the top of the backbone, we attach 2-layer MLP with a dimension of 2048-2048-128 as a projection head. For Higgs-100K, we set the $\tau = 0.1$, $\gamma = 1.2$, and $\alpha = 1/4096$. For CovType, we set $\tau = 0.2$, $\gamma = 1.1$, and $\alpha = 1/4096$. We train for 500 epochs with an SGD optimizer of momentum 0.9, base learning rate of 0.3, and the learning rate is scheduled by half-cycle cosine annealing. For evaluation of the CovType experiments, we train a linear classifier on the top of frozen representation for 100 epochs with a learning rate in $\{1, 3, 5\}$ and report the best validation accuracy (%). For evaluation of the Higgs-100K experiments, we use linear regression on the top of frozen representations by pseudo-inverse and report the validation accuracy (%). In Table 14, we provide the full experimental results on CovType and Higgs-100K. One can notice that the RM data augmentation is effective for the Higgs dataset, and FC data augmentation is effective for the CovType dataset.

## B.6 Graph experiments

For graph contrastive learning experiments on TUDataset [70], we follow the same experimental setup in [15]. Thus, we adopt the official implementation[6], and experiment over the hyper-parameter $\gamma \in \{1.5, 2.0, 3.0\}$ and choose the best parameter by evaluating on the validation dataset.

## C Mutual Information Estimation

### C.1 Implementation details

Assume we have $B$ batches of samples $\{x_i\}_{i=1}^B$ from $x_i \sim X$ and $B$ batches of samples $\{y_i\}_{i=1}^B$ from $y_i \sim Y$. Assume $(x_i, y_i) \sim P_{XY}$, i.e. positive pair, and $(x_i, y_j) \sim P_X P_Y$ for $i \neq j$. Then let $f^*(x, y)$ be optimal critic learned by maximizing CPC, MLCPC, or RMLCPC objectives. From the

---

[6]`https://github.com/Shen-Lab/GraphCL/tree/master/unsupervised_TU`

optimality condition, we have

$$f^*(x, y) \propto \log \frac{dP_{XY}(x, y)}{\alpha dP_{XY}(x, y) + (1 - \alpha)dP_X(x)dP_Y(y)} = \log \frac{r(x, y)}{\alpha r(x, y) + 1 - \alpha},$$

where $r(x, y) = \frac{dP_{XY}}{dP_X dP_Y}$ is a true density ratio. Then our idea is to recover $r$ by using $f^*$ and approximate the true mutual information. To do that, we first compute the log-normalization constant $Z$, i.e. $\frac{e^{f^*(x,y)}}{Z} = \frac{r(x,y)}{\alpha r(x,y)+1-\alpha}$. We use Monte-Carlo [44] methods by computing empirical mean as following:

$$\hat{Z} = \frac{\alpha}{B} \sum_{i=1}^{B} e^{f^*(x_i, y_i)} + \frac{1 - \alpha}{B(B - 1)} \sum_{i=1}^{B} \sum_{j \neq i} e^{f^*(x_i, y_j)}.$$

Then we have following approximate density ratio $\hat{r}$:

$$\hat{r}(x, y) = \frac{(1 - \alpha)e^{f^*(x, y)}}{\hat{Z} - \alpha e^{f^*(x, y)}}.$$

Since the mutual information is defined by the empirical mean of log-density ratio, we have

$$\hat{\mathcal{I}}(X; Y) = \frac{1}{B} \sum_{i=1}^{B} \log \hat{r}(x_i, y_i)$$

## C.2 MI estimation between correlated Gaussians

**Setup.** We follow the general procedure in [38, 24]. We sample a pair of vectors $(x, y) \in \mathbb{R}^{d \times d}$ from random variable $X \times Y$, where $X$ and $Y$ are unit normal distribution, and are correlated with correlation coefficient $\rho \in (0, 1)$. Then the true mutual information (MI) between $X$ and $Y$ can be computed by $\mathcal{I}(X; Y) = -\frac{d}{2} \log \left(1 - \frac{\rho}{2}\right)$. In our experiments, we set the initial mutual information to be 2, and we increase the MI by $\times 2$ for every 4K iterations. We consider joint critic, which concatenates the inputs $x, y$ and then passes through a 2-layer MLP with a dimension of $d - 256 - 1$, and the output of the fully connected layers are followed by the ReLU activation layer. For optimization, we use the Adam optimizer with learning rate of $10^{-3}$, $\beta_1 = 0.9$, $\beta_2 = 0.999$, batch size of 128.

**Results.** In Figure , we depict the curve of training curves and MI estimation that we proposed in Section 3.4. We consider $\alpha$-CPC, $\alpha$-MLCPC, $(\alpha, \gamma)$-RMLCPC with $\gamma = 2.0$. Remark that when $\alpha = 0.0$, the MI estimators exhibit large variance as we proved in Theorem 3.1. But for large enough $\alpha > 0$, the training objectives tend to have low variance, and thus we have a low variance MI estimator.

From Theorem 4.1, since there are total $128 \times 127$ negative pairs, we need to select $\alpha \propto 1/128$. Therefore, we divide the $\alpha$ by 128 for each experiment in Figure 2. One can observe that even though the optimal $\alpha$ varies across the objectives, choosing $\alpha$ around 1/128 gives a fairly good estimation of MI.

Note that the skew divergence is not only applicable to DV objectives, but also for the NWJ [79] objective, which is another variational lower bound of KL divergence defined as following:

$$\mathcal{I}_{\texttt{NWJ}}(f) := \mathbb{E}_{P_{XY}}[f(x, y)] - \mathbb{E}_{P_X P_Y}[e^{f(x,y)-1}],$$

and we have $\mathcal{I}(X; Y) = D_{\texttt{KL}}(P_{XY} \| P_X P_Y) = \sup_f \mathcal{I}_{\texttt{NWJ}}(f)$ [79]. Note that the NWJ objective is proven to have large variance as the true MI is too large [41]. However, we show that by introducing the NWJ objective for $\alpha$-skew KL divergence, one can lower the variance of the training objective, and perform MI estimation as we explained in Section 3.4. We refer the NWJ variational lower bound of $\alpha$-skew KL divergence as $\alpha$-NWJ, and is defined by following:

$$\mathcal{I}_{\texttt{NWJ}}^{(\alpha)}(f) := \mathbb{E}_{P_{XY}}[f(x, y)] - \alpha \mathbb{E}_{P_{XY}}[e^{f(x,y)-1}] - (1 - \alpha)\mathbb{E}_{P_X P_Y}[e^{f(x,y)-1}],$$

and we have $D_{\texttt{KL}}^{(\alpha)}(P_{XY} \| P_X P_Y) = \sup_f \mathcal{I}_{\texttt{NWJ}}^{(\alpha)}(f)$. Lastly, in Figure , we show that one can perform stable MI estimation through $\alpha$-NWJ objective by choosing appropriate $\alpha$.

**Algorithm 2** Pseudocode for MI estimation between the views

```
# aug: data augmentation that generates views
# x: batch of samples
# f: critic that takes a pair of inputs
# extract_pos_and_neg: a function that extracts positive and negative pairs (Alg. 1)

def MI_estimation(x, f, aug, alpha, gamma):
    x1, x2 = aug(x), aug(x) # two views
    pos, neg = extract_pos_and_neg(f(x1, x2)) # extract positive and negative pairs
    Z = alpha*pos.exp() + (1-alpha)*neg.exp().mean(dim=1) # normalization constant
    r_alpha = pos.exp() / Z
    r_true = (1-alpha)*r_alpha / (1-alpha*r_alpha)
    return r_true.log().mean()
```

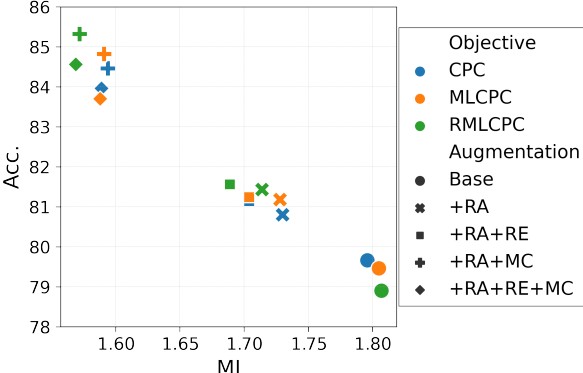

Figure 1: MI between the views and validation accuracy on ImageNet-100. RA: RandAugment [50], RE: RandomErasing [65], MC: Multi-crops [43].

### C.3   Mutual Information between the views and InfoMin principle.

**Implementation.**   We present the pseudocode for estimation of MI between the views in Algorithm 2. In Figure , we compute the MI for every iteration, and report the mean value of MI for the last epoch of training.

**InfoMin principle and RényiCL.**   Following [14], we plot the mutual information versus linear evaluation accuracy (%) by training with different data augmentations and contrastive objectives. In Figure 1, we observe that for harder data augmentations, the RMLCPC induces lower MI than that of CPC and MLCPC, while showing better downstream performances. This is because of the effect of easy positive and hard negative sampling explained in Section 4.3. Note that InfoMin [14] principle argues that the optimal views must share sufficient and minimal information about the downstream tasks. Here, one can observe that RényiCL intrinsically follows the InfoMin principle as it extracts minimal and sufficient information between the views. Therefore, RényiCL intrinsically follows InfoMin principle [14] that it extracts minimal and sufficient information between the views. Here, we establish the orthogonal relationship between the InfoMin principle [14] and RényiCL: InfoMin principle aims to find views that share minimal and sufficient information, while RényiCL can extract minimal and sufficient information from given views.

## D   Hyperparameter Sensitivity Analysis

The temperature is a hyperparameter that greatly affects the performance of contrastive learning [3, 86]. Here, we show that RMLCPC objective is robust to the choice of temperature hyperparameter. We conduct simple ablation studies on CIFAR-10 and CIFAR-100 with temperature $\tau = 0.5, 1.0$ and $\gamma = 1.5, 2.0$. We use the same experimental setup described in B.4 except the value of temperature. For comparison, we use same data augmentation setup; base data augmentation and hard augmentation using RandAugment and RandomErasing. In Table 15, we report the linear evaluation accuracy of each setting trained on CIFAR-10 and CIFAR-100 datasets. When using large temperature, i.e.,

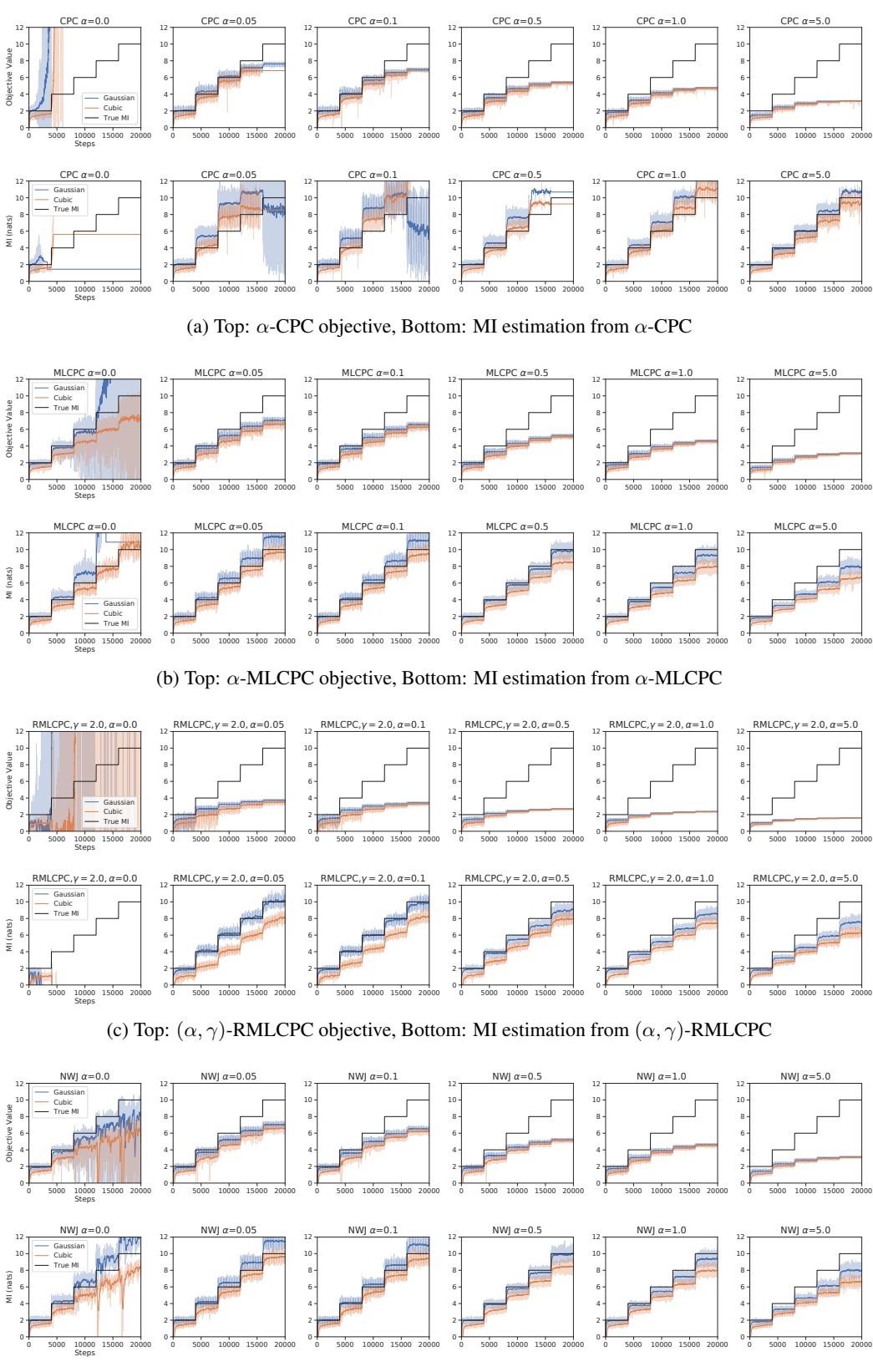

(a) Top: $\alpha$-CPC objective, Bottom: MI estimation from $\alpha$-CPC

(b) Top: $\alpha$-MLCPC objective, Bottom: MI estimation from $\alpha$-MLCPC

(c) Top: $(\alpha, \gamma)$-RMLCPC objective, Bottom: MI estimation from $(\alpha, \gamma)$-RMLCPC

(d) Top: $\alpha$-NWJ, Bottom: MI estimation from $\alpha$-NWJ

Figure 2: The plots for Gaussian MI estimation. We plot the values of training objectives, i.e., the negative of each objective, and the estimated MI value. For RMLCPC objective, we use $\gamma = 2.0$. Note that we divide the $\alpha$ by the batch size of 128.

$\tau = 1.0$, CPC and MLCPC degrades their performance when using harder data augmentation. On the other hand, RMLCPC consistently achieves better performance when using harder augmentations on both $\tau = 0.5$ and $\tau = 1.0$. Especially for CIFAR-100 dataset, using RMLCPC objective with $\tau = 1.0$ shows similar performance with using MLCPC objective with $\tau = 0.5$. Thus, our RényiCL does not only provide empirical gain, but also it could be a good starting point to search for effective hyperparameter due to their robustness to temperature.

Table 15: Sensitivity of CPC, MLCPC, RMLCPC on temperature. We report the linear evaluation accuracy (%) on CIFAR-10 and CIFAR-100 datasets. **Bold** entries denote the best performance.

| | CIFAR-10 | | | | CIFAR-100 | | | |
| | $\tau = 1.0$ | | $\tau = 0.5$ | | $\tau = 1.0$ | | $\tau = 0.5$ | |
| Method | Base | Hard | Base | Hard | Base | Hard | Base | Hard |
|---|---|---|---|---|---|---|---|---|
| CPC | 91.7 | 90.9 | 91.7 | 91.9 | 63.0 | 62.2 | 65.4 | 67.1 |
| MLCPC | 91.6 | 90.8 | 91.9 | 92.1 | 62.8 | 61.9 | 65.6 | 66.6 |
| RMLCPC ($\gamma = 1.5$) | 91.6 | 91.7 | 91.4 | **92.5** | 64.2 | 64.2 | 65.4 | 68.0 |
| RMLCPC ($\gamma = 2.0$) | 91.3 | 92.2 | 90.7 | **92.5** | 64.5 | 66.2 | 64.5 | **68.5** |