# OpenReview forum: "RényiCL: Contrastive Representation Learning with Skew Rényi Divergence"
_NeurIPS.cc/2022/Conference — NeurIPS 2022 Accept_

### Official Review · Reviewer_UaDs · 2022-07-11

**Rating:** 6
**Confidence:** 4
**Soundness:** 3 good
**Presentation:** 3 good
**Contribution:** 3 good

**Summary:**

This paper proposed the new robust contrastive learning scheme called RényiCL. By using the Rényi divergence instead of KL divergence as the objective function, the proposed method becomes robust against harder data augmentations. In order to keep the variance of the objective function small, this paper proposed the use of a variational lower bound for skewed Rényi divergence, which allows for training stabilization. The use of such skewed divergence is based on the observation that the original contrastive learning corresponds to the variational lower bound of the skewed KL divergence.

**Questions:**

## Implications of Theorem 3.2 for this study
The claim that motivates the technical contribution of this paper is Theorem 3.1, and the claim is roughly as follows: The estimated variance around the true solution of the Rényi divergence can be bounded from below by an exponential function on the difference between the KL divergence and the Rényi divergence. This observation indicates the variance of the naive objective could explode exponentially. Against this, Theorem 3.2 and additional theorem in the appendix argue that by introducing the skewed divergence instead of naive divergence, the estimated variance can be bounded from above by a constant. Although it attempts to show that exponential explosion of variance can be prevented because the constant appearing in the upper bound is independent of the number of samples, I believe this argument is inadequate for the following reasons

To begin with, the exponential function itself, which appears in the lower bound of the estimated variance, is also not a sample size dependent issue. The lower bound on the estimated variance in Theorem 3.1 depends essentially on $\gamma$ and the magnitude of divergence between distributions $P$ and $Q$, not on the number of samples. In other words, even if the constants appearing in the upper bound are independent of sample size, if they have exponential dependence on these variables, the upper bound is loose and meaningless in current context. The authors should explicitly mention in the text the order these constants have.

## Logical developments in the discussion of the gradient analysis for RényiCL
I have several concerns on the logical developments in the discussion on the gradient analysis. This paper discussed the effect of using $\alpha$-skewed Rényi divergence by investigating the gradient of the corresponding gradient. The brief logical flow is as follows:
1. Consider the case of $\alpha = 0$ for simplicity (the paper states it can be done without loss of generality)
2. Take the gradients for both baseline objective (MLCPC) and proposed Rényi objective (RMLCPC)
3. Construct the alternative objective functions of which gradient is equivalent to the original one
4. Interpret the implication of each term appears in the alternative objective function

I have two concerns about the above logical flow. The first one is a major and the second one is a minor concern.

### The simplification of $\alpha=0$ could mislead the discussion
By interpreting the positive first term for positive pairs and the negative second term for negative pairs, respectively, the authors state that the proposed method with $\gamma>1$ achieves easy positive sampling & hard negative sampling simultaneously compared to existing methods. However, it is only in the special case where $\alpha=0$ that terms for positive/negative pairs appear corresponding to exactly positive first/negative second terms, respectively. If we set α > 0 (which is done in practical experiment), the negative term which include positive samples will appears. For this reason, I believe that the claim that easy positive sampling occurs by using Rényi divergence with $\gamma>1$ somewhat ignores the real mechanism.

### The gradient of the alternative objective functions presented is not equivalent with original one
The paper states that the original objective function "is equivalent" to the alternative objective function in terms of gradient (e.g., p.6, l.184). However, this statement is not true **unless we introduce the stop-gradient operator for $q_{\theta}$**. This logical development itself is not a problem as long as the stop-gradient is mentioned, so the authors should clearly state how each equations related.

## Minor comments
- p.5, l.174: "$\alpha$-skew $\alpha$-skew Rényi divergence between ..." -> "$\alpha$-skew Rényi divergence between ..."

**Limitations:**

The authors have adequately addressed the limitations and potential negative societal impact of their work in conclusion section. In particular, the paper properly mentioned that it is difficult to identify what kind of data augmentation would be effective as a limitation of the proposed method. Yet, since the introduction of the paper motivates that the choice of data extension is the key in contrastive learning, there seems to be a slight discrepancy between the research objective and the actual proposed method.

**Strengths And Weaknesses:**

## Strength
- The flow of the text is clear, and the purpose, background, and proposed method of this study can be easily understood.
- The proposed method is simple to implement and can be easily used as a plug-in to existing SSL methods.
- Numerical experiments clearly demonstrate the effectiveness of the proposed method. The proposed method achieves comparable or slightly better performance than existing SSL methods when using normal data augmentation. In the situation with strong data augmentation, there was significant performance improvement over the existing methods, InfoMin and CLSA, for the two conditions of CUB200 and the Plant dataset.

## Weakness
- The significance of the theorem 3.2 is not adequately presented. This paper argues that skewed divergence allows the variance of the objective function to be bounded from above using some constants. However, this argument is not sufficiently convincing because there is not enough reference to the actual magnitude of the constant. More details are given in the Questions section
- Discussion about the gradient analysis of RényiCL is not fully convincing. The interpretation of the effect of the change in divergence is discussed by comparing the gradient of the objective function of RényiCL with the baseline gradient, but the conclusion is not convincing due to its logical development. I will discuss in detail in the Question section.
- Lack of ablation studies on robustness. Since the main concern of this paper is robustness to various data augmentation, the paper should directly demonstrate the effectiveness of the proposed method by comparing the performance by ablation of the contrastive objective under strong data augmentation.

---

> ### Author Response · Authors · 2022-08-02
> **Response to Reviewer UaDs (2 / 2)**
>
> __[C3] The simplification of $\alpha$=0 could mislead the discussion__
>
> __[A3]__ In Appendix B.1, we provide gradient analysis for the general $\alpha>0$ case. One can observe that the gradient of positive pairs in the second term is multiplied by $\alpha$, thus the gradient of positive pairs is affected by $\alpha$ and negative pairs.
> However, since we use a small value of $\alpha$, the gradient of the second term can be ignored in practice. To verify our claim, we experimented with different values of $\alpha$ on both basic and hard data augmentations:
>
> | $\alpha^{-1}$ |  1024 | 4096 | 16384 | 65536 |
> | ------------- |:----:|:----:|:----:|:----:|
> | Base Aug      | 79.0  | 79.3 | 78.6 | 78.4 |
> | Hard Aug      | 81.1  | 81.3 | 81.6 | 81.1 |
> | Gap              | 2.1    | 2.0   | 3.0   | 2.7   |
>
> One can observe that as $\alpha$ becomes smaller, the gap between using hard augmentation and the base augmentation becomes larger. Thus, there is a tradeoff in the choice of $\alpha$: the $\alpha$ should be large enough to evade large variance (Theorem 3.2), but small $\alpha$ is preferred to have the effect of easy positive sampling.
>
>
>
> __[C4] The validity of gradient analysis__
>
> __[A4]__ We apologize for the ambiguous explanation of the equivalent form in the gradient analysis. The stop gradient operator should be inserted in the importance weights. As you can see in our pseudocode in the appendix, we used the stop-gradient operator for the importance weights (i.e. torch.detach() function). In our revised manuscripts, we inserted the stop-gradient operator on importance weights to clarify the analysis. Thank you for pointing out our mistake!
>
> __[C5] Limitations of our work__
>
> __[A5]__ Remark that RényiCL achieves better performance by using the harder data augmentations. And not only RényiCL, but various self-supervised methods rely on the choice of data augmentation. In this paper, we did not search for an optimal data augmentation strategy, we simply used well-known hard data augmentations, e.g. RandAugment and Multi-crops.
>
> On the other hand, we believe that searching for a better data augmentation strategy is an interesting research objective. To clarify the limitations and scope of our work, we revised the Limitations and Future Works part in Section 6.

---

> > ### Comment · Reviewer_UaDs · 2022-08-08
> > **Response to Authors**
> >
> > Thank you for the detailed response.
> >
> > **[A1]** Additional ablation studies
> >
> > Thank you for providing additional ablation studies. These results addressed my concern on the empirical performance of the proposed method.
> >
> > **[A2]** Implication of Thm 3.2
> >
> > These additional analyses provided more detailed insight and understanding of the variance of $\alpha$-divergence. In particular, the fact that the variance bounce becomes loose when $\alpha$ is around 0 is considered to be consistent with the previous theorem (3.1), since $\alpha=0$ is the condition for recovering the original, non-skewed divergence.
> >
> > Since the reader may be concerned that other constants besides $c_2$ may not interfere with the main claim, I still believe that the order of all constants should be stated somewhere, even if the analysis shows that they are not relevant to the main claim.
> >
> > **[A3]** The simplification of $\alpha=0$ could mislead the discussion
> >
> > I was convinced by the responses given in this section and by the results of the experiment. Even if the gradient changes with a finite $\alpha$, it is reasonable to say that this is not a problem because we typically use smaller $\alpha$ in the experiments. However, a new (minor) concern arises here. If we are to use the smaller $\alpha$ for experiments, how would this be consistent with the Theorem 3.2 mentioned above? I was wondering if the $\alpha$ being tested in the experiments in this section is a sufficiently large value as required by Theorem 3.2. In other words, concerns remained about whether there is an $\alpha$ that can realistically satisfy both easy positive sampling and small estimated variance.
> >
> > **[Summary]**
> >
> > Most of the concerns raised in the initial review have been addressed. I still have the following two concerns, and would like to raise my score when these concerns are resolved.
> >
> > - I would like to know whether the constants other than $c_2$ in theorem 3.2 interfere with the main claim or not. If they are independent of $\alpha$ or of the divergence between $P$ and $Q$, it should be clearly stated so.
> > - Consistency of using a small $\alpha$ in experiments while theory requires a large enough $\alpha$ to suppress variance. Are these two requests practically feasible at the same time?

---

> > > ### Author Response · Authors · 2022-08-08
> > > **Response to Reviewer UaDs**
> > >
> > > We sincerely appreciate your additional feedback! In what follows, you can find our responses to your remaining two questions.
> > >
> > > __[C1] Other constants in Theorem 3.2__
> > >
> > > __[A1]__ Yes, the constants other than $c_2$ are independent of $\alpha$ or of $D_{\tt{KL}}(P \|\| Q)$, i.e., all of them are in order of $O(1)$. To clarify such potential concerns, we have elaborated the order of all constants in the revised manuscript. Thank you for your suggestion!
> > >
> > >
> > > __[C2] Tradeoff in choice of $\alpha$__
> > >
> > > __[A2]__ Here we have two different aspects in the choice of $\alpha$. If $\alpha$ is large,
> > > - (+) the estimator has low variance (Theorem 3.2), hence can stabilize the training
> > > - (-) the effect of easy positive sampling can be interfered.
> > >
> > > If $\alpha$ is small,
> > > - (+) the effect of easy positive sampling could be more utilized
> > > - (-) the training could be unstable
> > >
> > > Thus, there is a tradeoff in the choice of $\alpha$ for Rényi contrastive learning: training stability and effect of easy positive sampling. We empirically found a good choice of hyperparameter $\alpha$ for well-balancing the tradeoff.
> > >
> > > An idea to better circumvent the tradeoff is to use scheduling on $\alpha$ as Song et al. [1] did under the context of $\alpha$-MLCPC. The authors suggest using large $\alpha$ at the beginning of training (i.e., when training stability is more critical) and decaying the $\alpha$ throughout the training (i.e., as solution quality becomes more important). We also tried this scheduling at the early stage of our research, but we did not find significant gains; using a small, fixed value of $\alpha$ suffices. We do not make much effort to engineering $\alpha$, but we believe that our scheme can be further improved by carefully scheduling $\alpha$, which we leave for future work.
> > >
> > >
> > > __References__
> > >
> > > [1] Song, Jiaming, and Stefano Ermon. "Multi-label contrastive predictive coding." Advances in Neural Information Processing Systems 33 (2020): 8161-8173.

---

> > > > ### Comment · Reviewer_UaDs · 2022-08-08
> > > > **Reply**
> > > >
> > > > Thank you very much for the follow up comments.
> > > >
> > > > **[A1]**
> > > >
> > > > My concerns about the constants have been fully resolved. For me, this additional analysis makes the connection between Theorem 3.1 and 3.2 very clear.
> > > >
> > > > **[A2]**
> > > >
> > > > I clearly understood that there is a trade-off between an easy positive sampling effect and numerical stability. I also agree that experimentally, a small $\alpha$ can promote easy positive sampling and achieve good performance. However, my concern is whether Theorem 3.2 is really a reasonable theory to explain the experimental phenomenon in that case. For example, the paper mentioned that $\alpha = 1/65,536$ is used in the ImageNet experiments as in Appendix B.3. Since $c\_2(\alpha) = \min \\{ \frac{1}{\alpha}, \frac{ \chi^2(P||Q) }{ 1 - \alpha } \\}$, the upper bound of the estimator variance should be at least greater than $65,536$ for the case $P$ and $Q$ have an adequate distance. I consider this a sufficiently likely situation as long as $P$ and $Q$ are not the same distribution. There remains a concern as to whether an estimated variance of the order of $1e5$ can be said to provide a practically tight bound compared to the estimator variance for the divergence using original non-skew distribution.

---

> > > > > ### Author Response · Authors · 2022-08-08
> > > > > **Response to Reviewer UaDs**
> > > > >
> > > > > Thank you for your insightful comments! First of all, we would like to highlight the importance of Theorem 3.2. This is the first result which provides a theoretical guarantee on how CPC and MLCPC have low variance. This theoretical guarantee indeed motivated us to propose the Rényi MLCPC objective, which is a low variance skew Rényi divergence estimator that allows contrastive learning with Rényi divergence.
> > > > >
> > > > > Theorem 3.2 also  provides an upper bound for contrastive objectives without assuming a specific form of neural critic $f$, while in the case of contrastive ``representation’’ learning, we use the following cosine similarity critic:
> > > > > $$
> > > > > f_{\tt{cos}}(x,y) = \frac{\langle h(x), h(y) \rangle}{\tau \|\|h(x)\|\| \|\|h(y)\|\|},
> > > > > $$
> > > > > where $\tau$ is a temperature. Hence, Theorem 3.2 may be not tight for $f_{\tt{cos}}$. To be specific, since $f_{\tt{cos}}$ is already a bounded function, one can derive an alternative upper bound (with respect to $\tau$) on the variance of the KL divergence estimator as done in [1]; intuitively, when using a large value of $\tau$, one can lower the variance. Thus, for the cosine similarity critic, one can derive a tighter upper bound on the variance of skew KL divergence estimator that involves $\tau$. Then, we believe that estimation with cosine similarity critic allows a smaller value of $\alpha$ which we empirically observed in ImageNet experiments. While investigating a tighter bound for specific $f$ is an interesting topic, however, our work aims to demonstrate the general purpose analysis for CPC and MLCPC, which we further generalize to the Rényi MLCPC case.
> > > > >
> > > > > [1] Song, Jiaming, and Stefano Ermon. "Understanding the limitations of variational mutual information estimators." arXiv preprint arXiv:1910.06222 (2019).

---

> > > > > > ### Comment · Reviewer_UaDs · 2022-08-09
> > > > > > **Response to Authors**
> > > > > >
> > > > > > Thanks to the authors for a clear explanation.
> > > > > >
> > > > > > Now I understood the difference/contribution of Theorem 3.2 from similar existing results. I agree that it is worthwhile to bound the estimated variance without specifying the specific form of the critic function. I also understood that we could compose a more practical tight bound under a given specific critic function. Since all concerns have been addressed, I would like to raise my score to 6.

---

> > > > > > > ### Author Response · Authors · 2022-08-09
> > > > > > > **Response to Reviewer UaDs**
> > > > > > >
> > > > > > > Great! We sincerely appreciate the valuable discussions throughout the rebuttal period, it really helped strengthening our paper.
> > > > > > >
> > > > > > > Meanwhile, we are still open to discussion about any aspect of our paper. Please let us know if there is any further questions.
> > > > > > >
> > > > > > > Thank you,
> > > > > > >
> > > > > > > Paper 10007 Authors

---

> ### Author Response · Authors · 2022-08-02
> **Response to Reviewer UaDs (1 / 2)**
>
> We sincerely appreciate your thoughtful comments, efforts, and time. We respond to each of your questions and concerns one-by-one in what follows. We also kindly ask you to check out the revised manuscript we have updated together. Please let us know if you have any comments/concerns that we have not addressed up to your satisfaction.
>
> __[C1] Robustness of RényiCL to the data augmentation__
>
> __[A1]__ Throughout the paper, we claim that RényiCL is beneficial when incorporated with harder data augmentations. Since you asked about the robustness of RényiCL under various data augmentations, we perform additional experiments on the robustness of RényiCL on various data augmentation schemes.
>
> We first show that RényiCL is more robust to the noisy data augmentation, as similar to that in [1]. To that end, we apply Noisy Crop, which additionally performs RandomResizedCrop of size 0.2.
> Then the generated views might contain only nuisance information (e.g. background), thus it can hurt the generalization of contrastive learning.
> We train each model with CPC, MLCPC, and RMLCPC objectives and data augmentations with and without Noisy Crop. We experimented on CIFAR-10, CIFAR-100, and ImageNet-100 datasets, and the results are shown below.
>
> | Method | CIFAR10    | CIFAR100  | ImageNet100 |
> | ------ | :-------:  | :------:  | :---------: |
> | CPC    | 91.7       | 65.4      | 79.7     |
> | CPC w/ NC    | 90.6(-1.1) | 64.9(-0.5)| 79.2(-0.5)|
> | MLCPC  | 91.9       | 65.6      | 79.5        |
> |MLCPC w/ NC     | 90.7(-1.2) | 65.0(-0.6)| 79.4(-0.1)  |
> |RMLCPC  | 90.7       | 64.5      | 78.9        |
> |RMLCPC w/ NC     | __91.6(+0.9)__ | __67.6(+3.1)__| __80.5(+1.6)__  |
>
> One can observe that while the performance of using CPC and MLCPC degrades as we apply the noisy crop, RMLCPC conversely improves the performance. Hence, RMLCPC is robust in the choice of data augmentation.
>
> Second, we experiment when there is limited data augmentation. For each CIFAR-10, CIFAR-100, and STL-10 dataset, we remove data augmentations such as color jittering and grayscale and only apply RandomResizedCrop and HorizontalFlip. The results are shown below:
>
> | Method | CIFAR10    | CIFAR100  | STL10 |
> | ------ | :-----:    | :------:  | :---------: |
> | CPC    | 63.2       | 31.8      | 52.8        |
> | MLCPC  | 63.3       | 32.9      | 53.1        |
> |RMLCPC  | __67.1__   | __41.2__  | __56.4__    |
>
> Note that RMLCPC achieves the best performance among CPC and MLCPC in the case of limited data augmentation. Therefore the RMLCPC objective is not only robust to the hard data augmentation but is also robust when there is limited data augmentation.
>
> __References__
>
> [1] Chuang, Ching-Yao, et al. "Robust contrastive learning against noisy views." Proceedings of the IEEE/CVF Conference on Computer Vision and Pattern Recognition. 2022.
>
>
> __[C2] Implication of Theorem 3.2__
>
> __[A2]__ Remark that Theorem 3.1 demonstrates the asymptotic lower bound to the variance of variational Rényi divergence estimator, which is exponentially large with respect to the true KL divergence. Thus, Theorem 3.1 implies that one requires $n=\Theta\left(e^{D_{\tt{KL}}(P \|\|Q)}\right)$ to achieve low variance.
>
> On the other hand, Theorem 3.2 states that the variance of $\alpha$-skew KL divergence is asymptotically bounded if we use sufficiently large $\alpha$. For the case when $\alpha$=0 or small $\alpha\approx 0$, the variance of variational $\alpha$-skew KL divergence estimator can be prohibitively large if we don’t have sufficiently many samples. Indeed, regarding Reviewer UaDs’s question, we further analyze the constants in Theorem 3.2. Remark that the constant $c_2$ in Theorem 3.2 is bounded by both $\frac{1}{\alpha}$ and $\frac{\chi^2(P\|\|Q)}{1-\alpha}$ (which we added proof in the appendix of revised manuscript),
> and remark that $\chi^2$ divergence satisfies $\chi^2(P\|\|Q) \geq e^{D_{\tt{KL}}(P\|\|Q)}-1$.
> Thus, if $\alpha$ is too small that $\frac{1}{\alpha}$ is bigger than $\frac{\chi^2(P\|\|Q)}{1-\alpha}$, the bound in Theorem 3.2 becomes loose as it has the same order as Theorem 3.1 so that it still suffers from large variance. Therefore, Theorem 3.2 does not always guarantee bounded variance for any $\alpha$: our analysis implies that one should choose large enough $\alpha$ to avoid large variance.

---

### Official Review · Reviewer_dj8j · 2022-07-11

**Rating:** 8
**Confidence:** 5
**Soundness:** 4 excellent
**Presentation:** 4 excellent
**Contribution:** 3 good

**Summary:**

This paper investigates a novel alternative formulation of contrastive learning that leverages the Renyi divergence. Motivated by the sensitivity of augmentation schemes for the learned representations, the author(s) argues using Renyi divergence can better manage hard augmentations. To overcome the variance issue in the naive formulation, a more stable version based on skewed statistics is advocated, with theoretical justifications. The author(s) showed that using the proposed RenyiRL can spared the need for additional hacks or extra computations to get decent performance. Strong empirical evidence supports the claims made in the paper.

**Questions:**

See my first comment in the Weakness section.

**Ethics Review Area:**

["I don’t know"]

**Limitations:**

I don't have any concerns on limitations and potential negative societal impact.

**Strengths And Weaknesses:**

### Strength

* *(Quality, Originality)* The application of Renyi contrastive learning for self-supervised learning is novel. Although I have seen other recent works also attempting to Renyi for contrastive learning, I believe this manuscript is the only one that well balances theoretical and practical aspects.
*  *(Clarity, Significance)* This paper is written with exceptional clarity and I enjoyed reading it. It provides an extensive coverage on related work, and I am genuinely surprised to see some research threads not well known to the general contrastive learning community. Nice work. I added some additional thread which should make the literature review more comprehensive, and will hopefully inspire future research ideas.
* *(Quality)* Strong experimental results, well curated experimental code.

### Weakness

* It is not clear to me how the variance in Theorem 3.2 turns into the variance in Theorem 3.1 in the limiting case $\alpha=0$, could the author(s) elaborate on that?

* Variance is not the only reason causing InfoNCE to perform sub-optimally. Discussions from [5-7] have presented diverse perspectives on how contrastive learning can be improved based numerical or optimization views, also backed up strong mathematical arguments. These works are complementary to the approach adopted here and should be discussed.

### Misc

* Theorem 3.1 is related to the result from [1] on the exponential variance issue for MI estimation, which should be  discussed.
* The objective function for skewed $\chi^2$ divergence presented in the [Tsai et al. 2021] (which this work is based on) is related to the spectrum contrastive learning objective [5], which is simpler and also demonstrated strong performance gains over the vanilla InfoNCE objective. The Renyi divergence and $\chi^2$ divergence have also been successfully applied in the context of generative modeling [2-4], where the goal of minimizing $D_{\alpha}(P_{XY}\parallel P_XP_Y)$ or $D_{\alpha}(P_X\parallel Q_X)$ is shared.
* Potential typos: $\alpha$-CPC is actually proposed by [B Poole et al. ICML 2019], not [J Song, et al. NeurIPS 2020].

[1] McAllester, D. and Stratos, K. Formal limitations on the measurement of mutual information. AISTATS 2020
[2] Y Li, et al. Rényi Divergence Variational Inference. NeurIPS 2016
[3] L Chen, et al. Variational inference and model selection with generalized evidence bounds. ICML 2018
[4] C Tao. Chi-square Generative Adversarial Network. ICML 2018
[5] J HaoChen, et al. Provable Guarantees for Self-Supervised Deep Learning with Spectral Contrastive Loss. NeurIPS 2021
[6] Q Guo, et al. Tight Mutual Information Estimation With Contrastive Fenchel-Legendre Optimization. 2021
[7] J Chen, et al. Simpler, Faster, Stronger: Breaking The log-K Curse On Contrastive Learners With FlatNCE. 2021

---

> ### Author Response · Authors · 2022-08-02
> **Response to Reviewer dj8j**
>
> We sincerely appreciate your thoughtful comments, efforts, and time. We respond to each of your questions and concerns one-by-one in what follows. We also kindly ask you to check out the revised manuscript we have updated together. Please let us know if you have any comments/concerns that we have not addressed up to your satisfaction.
>
> __[C1] About the relationship between Theorem 3.1 and Theorem 3.2__
>
> __[A1]__ Remark that Theorem 3.1 demonstrates the asymptotic lower bound to the variance of variational Rényi divergence estimator, which is exponentially large with respect to the true KL divergence. Thus, Theorem 3.1 implies that one requires $n=\Theta\left(e^{D_{\tt{KL}}(P \|\|Q)}\right)$ to achieve low variance.
>
> On the other hand, Theorem 3.2 states that the variance of $\alpha$-skew KL divergence is asymptotically bounded if we use sufficiently large $\alpha$. For the case when $\alpha$=0 or small $\alpha\approx 0$, the variance of variational $\alpha$-skew KL divergence estimator can be prohibitively large if we don’t have sufficiently many samples. Indeed, regarding Reviewer UaDs’s question, we further analyze the constants in Theorem 3.2. Remark that the constant $c_2$ in Theorem 3.2 is bounded by both $\frac{1}{\alpha}$ and $\frac{\chi^2(P\|\|Q)}{1-\alpha}$ (which we added proof in the appendix of revised manuscript),
> and remark that $\chi^2$ divergence satisfies $\chi^2(P\|\|Q) \geq e^{D_{\tt{KL}}(P\|\|Q)}-1$.
> Thus, if $\alpha$ is too small that $\frac{1}{\alpha}$ is bigger than $\frac{\chi^2(P\|\|Q)}{1-\alpha}$, the bound in Theorem 3.2 becomes loose as it has the same order as Theorem 3.1 so that it still suffers from large variance. Therefore, Theorem 3.2 does not always guarantee bounded variance for any $\alpha$: our analysis implies that one should choose large enough $\alpha$ to avoid large variance.
>
>
> __[C2] On various perspectives on contrastive learning__
>
> __[A2]__ We sincerely appreciate your comments on the various perspectives in contrastive learning. While we focus on the innate easy positive and hard negative importance sampling process of RényiCL to explain the empirical gain, we believe that other components can be complementary to our method. For example, it would be nice if we can derive a similar theoretical guarantee for the linear probing of representations trained by RényiCL as done in [1]. Moreover, the idea of FlatNCE [2] or Fenchel Legendre transform [3] could be investigated under the Rényi divergence scheme which is an interesting future direction. Lastly, we notice that the gradient analysis in our work is related to that in [3], so we add references for it. Again, thank you for the insightful comments and references, and those things are updated in the Conclusion section of our revised manuscripts!
>
> __References__
>
> [1] HaoChen, Jeff Z., et al. "Provable guarantees for self-supervised deep learning with spectral contrastive loss." Advances in Neural Information Processing Systems 34 (2021): 5000-5011.
>
> [2] Chen, Junya, et al. "Simpler, faster, stronger: Breaking the log-k curse on contrastive learners with flatnce." arXiv preprint arXiv:2107.01152 (2021).
>
> [3] Guo, Qing, et al. "Tight mutual information estimation with contrastive Fenchel-Legendre optimization." arXiv preprint arXiv:2107.01131 (2021).

---

> > ### Comment · Reviewer_dj8j · 2022-08-08
> > **Thanks for the rebuttal**
> >
> > I have carefully read the author(s)' rebuttal and their discussions with other reviewers. My confusion has been clarified and I also learned some interesting perspectives from the author-reviewer discussions. I am also impressed with the fact that the author(s) added more empirical results given the short time allowed by the rebuttal phrase. This is a very strong work and I enthusiastically recommend it for acceptance.

---

> > > ### Author Response · Authors · 2022-08-09
> > > **Response to Reviewer dj8j**
> > >
> > > Thank you for your positive response! We are happy that our further empirical results and analyses helped to address your concerns and provided some interesting perspectives.
> > >
> > > If you have any further questions or concerns, please do not hesitate to let us know.
> > >
> > > Thanks,
> > >
> > > Paper 10007 Authors

---

### Official Review · Reviewer_6o5u · 2022-07-12

**Rating:** 7
**Confidence:** 4
**Soundness:** 4 excellent
**Presentation:** 4 excellent
**Contribution:** 3 good

**Summary:**

This paper revisits the derivations of the InfoNCE objective from the original *Contrastive Predictive Coding* paper, replacing the KL divergence in the mutual mutual information with the Renyi divergence. As with the KL divergence, the Renyi divergence is known to enjoy a variational representation, which in both cases has a naive estimator that suffers from high variance. They then note a variance reduction idea, which I do not recognize from the contrastive literature, that shows that for D \in \{KL, Renyi\}, replacing D(P|Q) with D(P | alpha*P + (1-alpha)*Q) for some small alpha yields an estimator with much improved variance. The paper proposes to use this variance reduction with the Renyi divergence as a plug-in replacement for InfoNCE as a training objective. The final theoretical component is to show that the resulting objective has gradients that naturally place high importance weights on hard negative and easy positives. The authors suggest that the easy positive in particular lend themselves to using more aggressive data augmentations in practice. Experiments cover visual, tabular, and graph structured data.

**Questions:**

See previous box, particularly regarding experimental comparisons.

**Limitations:**

Yes.

**Strengths And Weaknesses:**

Congratulations to the authors on a well written, tidy piece of work. There are certainly some interesting ideas in this work, and I enjoyed reading this paper. First I outline some of the positive aspects I saw in this work, before turning to some weaker points and a few concerns I have, particularly with some of the experimental comparisons.

It is good to see that the thought process behind the mutual information —> InfoNCE derivation can be replicated for other divergences. Perhaps the same could be done with all sorts of other divergences, leading to a variety of different behaviors in objective functions. I also liked the presentation of the skew-divergence connection. The skew-divergence seems like it has useful low-variance, high-bias properties that could be useful in other contrastive learning contexts.

One of the premier benefits of RenyiCL is touted to be able to gracefully handle, and make use of, aggressive positive sampling thanks to its weighting biassing towards easy positives. As such the authors use what is said to be “harder data augmentations” for their method.  One of the ingredients constituting “harder augmentations” is multi-crop sampling à la SwAV. This raises a major concern of a lack of apples to apples vision comparisons, since the multi-crop method has, in all cases I have ever seen, always boosted performance, and increase the memory and runtime requirements. So the results reported in Tables {1,2,3} all compare a method using multi-crop to methods that do not (except for SwAV, which does and performs similarly to RenyiCL). It would be much fairer to remove the multi-crop sampling from RenyiCL. Table 4 has this number, but it is only for a 200 epoch run so again it is not really fair to compare. Finally, Table 5 compares   RenyiCL to other methods using the hard data augmentations. An important bright-spot here is that RenyiCL does see the biggest boost of the three methods, which is great to see. However RenyiCL is only compared to CPC and a variant of CPC; it remains unclear, for instance, how MoCo-v3 would fare in comparison using multi-crop. In all, it remains hard to say exactly how RenyiCL compares to other methods, besides to say that it is likely broadly comparable.

The inclusion of experiments on tabular and graph data is welcome, and I am glad to see promising results on other domains besides vision.

I end with a much more philosophical point. The motivation for “why Renyi” divergence is mostly post hoc—instead of any intrinsic explanation for why Renyi may be a better estimator than KL, the authors first go through the process of deriving their low-variance estimator, and analyzing its gradients. They then find a number of natural, potentially desirable, attributes of the final objective such as easy positive/hard negative sample importance weights.  This is, again, very similar to process of producing the InfoNCE objective, starting from mutual information. It was convincingly argued by Tschannen & Djolonga et al. (2020), that it is the form of the InfoNCE loss itself, rather than any relation to mutual information, that explains its success. I have to say that I very much expect the same to hold for the-based Renyi objective presented in this work. From this point of view, it is worth putting to one side for a moment all of the scaffolding built on top of the idea of the Renyi divergence, and simply focus on the form of the final RenyiCL objective. As noted by the authors, this objective is of similar form to prior idea, but with an importance sampling re-weighting to emphasize hard negatives and easy positives. The main distinction here from prior works, which have considered hard negatives and hard positives using similar importance weightings, is to keep using hard negatives, but instead to reverse the focus to be on *easy positives*, and note that actually this may buy some benefits in terms of being able to use more aggressive data augmentations. This is an interesting proposal, and may well be at the core of the effectiveness of the proposed method; the Renyi divergence may mostly serve to obfuscate this.


To conclude, while I have some gripes with the vision experimental setup, and suspect that the main benefit of RenyiCL of easy positive sampling could similarly be implemented in any other contrastive framework using importance weights, I am still broadly supportive of this work. To me this core recipe of easy positive weights + aggressive data augmentations (and its justification in Table 5) is already an interesting contribution, since how to produce good positive samples is perhaps the single most important question in contrastive learning. On a different note, the paper also offers other ideas, such as the skew-divergences, and their applicability to both KL and Renyi divergences seem interesting and may be of independent interest for producing low-variance contrastive learning objectives.

----

Post rebuttal: increased score by one.

---

> ### Author Response · Authors · 2022-08-02
> **Response to Reviewer 6o5u (2 / 2)**
>
> __[C2] Motivation for Rényi contrastive learning__
>
> __[A1]__ The Rényi divergence is a generalization of KL divergence that is defined with an additional parameter $\gamma$ called an order. As the order $\gamma$ becomes higher, the Rényi divergence penalizes more when two distributions misalign. Therefore, since the goal of contrastive representation learning is to train an encoder to discriminate between positives and negatives, we hypothesize that maximizing the Rényi divergence between positives and negatives could lead to more discriminative representations when the data augmentations are aggressive. This is indeed confirmed by our extensive experiments. Furthermore, our theoretical analysis shows that Rényi contrastive learning weighs importance on easy positives and hard negatives by using a higher order $\gamma$, which provides an additional benefit using the Rényi divergence.
>
> While easy positive and hard negative importance sampling for contrastive learning is one of the main contributions of our work, we believe that our paper presents more general ideas on contrastive learning objectives: generalized contrastive learning by variational estimation of skew Rényi divergences. Thus, we believe that this concept could be applied to various machine learning tasks other than contrastive representation learning such as Information Bottleneck as we mentioned in the last section.
>
> Finally, to clarify our messages, we polish the writing to decouple our contribution into two parts: first, we explain the proposed variational skew Rényi divergence estimator, and second, we propose Rényi contrastive learning by using proposed objectives and show how it performs easy positives and hard negatives sampling. We hope this change would enhance the readability and clarify our contributions.

---

> > ### Comment · Reviewer_6o5u · 2022-08-08
> > **Response to rebuttal**
> >
> > Thank you for the detailed response. The extra clarifying experiments are appreciated (though I should say I never expect authors to run new experiments during rebuttal!). I am happy with all the details provided and continue to think this is a good piece of work, and increase my score to signify this.

---

> > > ### Author Response · Authors · 2022-08-09
> > > **Response to Reviewer 6o5u**
> > >
> > > Thank you for the positive response! It is great that our new experimental results addressed your concerns.
> > >
> > > If you have any further questions or concerns, pleast let us know.
> > >
> > > Thanks.
> > >
> > > Paper 10007 Authors.

---

> ### Author Response · Authors · 2022-08-02
> **Response to Reviewer 6o5u (1 / 2)**
>
> We sincerely appreciate your thoughtful comments, efforts, and time. We respond to each of your questions and concerns one-by-one in what follows. We also kindly ask you to check out the revised manuscript we have updated together. Please let us know if you have any comments/concerns that we have not addressed up to your satisfaction.
>
> __[C1] Fair comparison with other contrastive learning methods__
>
> __[A1]__ We claim that Renyi contrastive learning shows better performance in visual self-supervised representation learning with the usage of harder data augmentation such as multi-crop and RandAugment. However, you mentioned that the comparison with other self-supervised methods should be done under the same data augmentation because data augmentations like multi-crop are mostly helpful for various self-supervised methods.
>
> To address your concern, we conduct additional experiments on MoCo v3 [1] and DINO [2], which are state-of-the-art methods of visual self-supervised learning. We first reimplemented MoCo v3 with a smaller batch size (original 4096 to 1024) and in our setting, we achieved 69.6%, which is better than their original reports (68.9%). Then, we use additional RandAugment (RA), RandomErasing (RE), and multi-crop (MC) when training MoCo v3. Furthermore, we compare RényiCL with DINO. Note that DINO uses default multi-crops, therefore we further applied RandAugment and RandomErasing for a fair comparison. Remark that all results are run by us with 100 training epochs.
>
> |Data Aug. | MoCo v3 [1] | DINO [2] | __RényiCL__ |
> |:------| :-----:| :-:| :--:|
> | Base  | 69.6   | 70.9 | 69.4 |
> | Hard  | 73.5    | 72.5 | __74.3__ |
> | Gain  | +3.9    | +1.6 | __+4.9__  |
>
> One can observe that while MoCo v3 and DINO benefit from using harder data augmentation, RenyiCL shows the best performance when using harder data augmentation. Also, compared to the Base data augmentation, RényiCL attains the most gain by using harder data augmentation.
>
> Finally, we would like to highlight the effectiveness of RenyiCL by showing extra results. Remark that RenyiCL achieved 75.3% on ImageNet linear probing with 200 epochs of training. We found out that RenyiCL achieves 76.2% with 300 epochs of training, which outperforms state-of-the-art methods, which trained for very long epochs, by a large margin:
>
> | Method   | Epochs | Linear Eval. |
> | :--------   | :-------: | :------------: |
> | BYOL [5] | 1000     | 74.3 |
> | MoCo v3 [1] | 1000   | 74.8      |
> | DINO [2]  | 800    | 75.3      |
> | SwAV [6] | 800    | 75.3      |
> | NNCLR [3]  | 1000   | 75.6      |
> | C-BYOL [4] | 1000   | 75.6      |
> | __RényiCL__| __300__    | __76.2__      |
>
>
> __References__
>
> [1] Chen, Xinlei, Saining Xie, and Kaiming He. "An empirical study of training self-supervised vision transformers." Proceedings of the IEEE/CVF International Conference on Computer Vision. 2021.
>
> [2] Caron, Mathilde, et al. "Emerging properties in self-supervised vision transformers." Proceedings of the IEEE/CVF International Conference on Computer Vision. 2021.
>
> [3] Dwibedi, Debidatta, et al. "With a little help from my friends: Nearest-neighbor contrastive learning of visual representations." Proceedings of the IEEE/CVF International Conference on Computer Vision. 2021.
>
> [4] ​​Lee, Kuang-Huei, et al. "Compressive visual representations." Advances in Neural Information Processing Systems 34 (2021): 19538-19552.
>
> [5] Grill, Jean-Bastien, et al. "Bootstrap your own latent-a new approach to self-supervised learning." Advances in neural information processing systems 33 (2020): 21271-21284.
>
> [6] Caron, Mathilde, et al. "Unsupervised learning of visual features by contrasting cluster assignments." Advances in Neural Information Processing Systems 33 (2020): 9912-9924.

---

### Official Review · Reviewer_B1D9 · 2022-07-12

**Rating:** 5
**Confidence:** 2
**Soundness:** 2 fair
**Presentation:** 2 fair
**Contribution:** 2 fair

**Summary:**

The paper proposes to use the variational estimation of the (lower bound of the) skew renyi divergence for contrastive representation learning. The authors compare the reynyi divergence against traditional divergence such as KL divergence and skew KL divergence, and argue the proposed skew renyi divergence is beneficial for contrastive representation learning empirically.

**Questions:**

1. equation below line 86: g is not defined
2. issues with using Donsker style variational estimation: The sup is over all possible functions. However in practice one has to assume some function class which is more restrictive. In the end the estimation is only a lower bound.
3. line 212: can we have some details on how to estimate the constant Z using Monte-Carlo methods? how complex is that? usually this is a headache for a lot of problems



**Strengths And Weaknesses:**

While it is nice to have a new divergence and understand some properties of it, I am not quite convinced why the proposed skew Renyi divergence is better than the traditional ones, for example, skew KL divergence. Besides the empirical results, is there any explanation why skew Renyi divergence is preferred?

Another thing that bothers me is the paper entangles the analysis of of the skew Renyi divergence with data augmentation. I am curious if it is possible to decouple the two. One can do contrastive learning using the estimator of the skew Renyi divergence without data augmentation. Coupling the two makes the analysis harder to read. In particular, I don't feel the transition probability A(v|x) is well explained. Why do authors need to use the transition probability to define the positive and negative pairs? Can we define them just using the data distribution without the transition probability?

---

> ### Author Response · Authors · 2022-08-02
> **Response to Reviewer B1D9**
>
> We sincerely appreciate your thoughtful comments, efforts, and time. We respond to each of your questions and concerns one-by-one in what follows. We also kindly ask you to check out the revised manuscript we have updated together. Please let us know if you have any comments/concerns that we have not addressed up to your satisfaction.
>
> __[C1] Reasons for using skew Renyi divergence__
>
> __[A1]__ The Rényi divergence is a generalization of KL divergence that is defined with an additional parameter $\gamma$ called an order. As the order $\gamma$ becomes higher, the Rényi divergence penalizes more when two distributions misalign. Therefore, since the goal of contrastive representation learning is to train an encoder to discriminate between positives and negatives, we hypothesize that maximizing the Renyi divergence between positives and negatives could lead to more discriminative representations when the data augmentations are aggressive. This is indeed confirmed by our extensive experiments. Furthermore, our theoretical analysis shows that Renyi contrastive learning weighs importance on easy positives and hard negatives by using a higher order $\gamma$, which provides an additional benefit using the Renyi divergence.
>
> __[C2] Data augmentation and Renyi contrastive learning__
>
> __[A2]__ To clarify your concern, the proposed Renyi contrastive learning objective, and data augmentation are independent components. Also, one can use Renyi contrastive learning without data augmentation. Hence, following your suggestion, we significantly revised the structure of the paper in our new draft; we first explain the proposed variational skew Renyi divergence estimator and then the application of the proposed estimator to the contrastive representation learning with data augmentations. We do believe that this decoupling would increase readability significantly. Thank you very much for the suggestion!
>
> __[C3] Transition probability and positive/negative pairs__
>
> __[A3]__ We defined the transition probability A(v|x) to formalize the concept of positive and negative pairs in contrastive learning. However, we found out that the explanation can hurt readability. In our revised manuscript, we polish our writing and present contrastive representation learning with much simpler notation.
>
> __[Q4] Equation below line 86: g is not defined__
>
> __[A4]__ At the beginning of the paragraph, we define $g$ to be an encoder that takes data and outputs features of the lower dimension that contains the information of the data. Nevertheless, to provide better readability, we insert the information on $g$ for each equation.
>
> __[Q5] Issues with Donsker style variational estimation__
>
> __[A5]__ While we use the restricted class of functions in variational estimation, we use deep neural networks (DNN) for the estimation which guarantees the universal approximation. In practice, previous works such as MINE [1], showed that using DNN suffices in many applications such as mutual information estimation and generative modeling. In Appendix D, experiments on the estimation of mutual information between synthetic Gaussian datasets show that our neural networks can estimate the mutual information with low bias and low variance.
>
> Finally, we would like to emphasize that our goal is not to exactly estimate the probability divergences between two distributions, but rather, to train a representation by maximizing probabilistic diverges. This is the reason why we resort to the variational method when measuring probabilistic divergences. To narrow the main scope of our paper, we defer the mutual information parts to the appendix.
>
> __[Q6] On Monte-Carlo methods__
>
> __[A6]__ In the mutual information estimation part, we approximate the log-normalization constant $Z$ by simply computing the empirical mean:
> $$
> \hat{Z} = \frac{\alpha}{B}\sum_{i=1}^B e^{f(x_i,y_i^+)} + \frac{1-\alpha}{B(B-1)} \sum_{i=1}^B \sum_{j\neq i} e^{f(x_i, y_{ij}^-)}
> $$
> We found that this simple Monte-Carlo method suffices in our Gaussian mutual information task, while we believe that one can use better strategies to compute the log-normalization constant. We leave it for future work.
>
>
> __References__
>
> [1] Belghazi, Mohamed Ishmael, et al. "Mutual information neural estimation." International conference on machine learning. PMLR, 2018.

---

### Author Response · Authors · 2022-08-02
**Official common response**

Dear reviewers and AC,

We sincerely appreciate your thoughtful comments, efforts, and time in reviewing our manuscript.

As reviewers highlighted, we believe our paper is well-written (6o5u, dj8j, UaDs) and provides extensive coverage (dj8j) with strong empirical evidence supporting our claim (dj8j, UaDs). In particular, our contributions are following:
- We propose a novel Rényi contrastive learning objective that performs variational estimation of skew Rényi divergence.
- We show that Rényi contrastive learning performs easy positive and hard negative sampling, which is useful to handle harder data augmentations.
- Especially, our method achieves SOTA 76.2% under ImageNet self-supervised classification on ResNet-50 with only 300 epochs of training.

We appreciate your incisive comments on our manuscript. In response to the questions and concerns you raised, we have carefully revised and improved the manuscript with the following additional discussions and experiments:
 - Revised the method section into variational estimation part (Section 3) and contrastive learning part (Section 4) to decouple Rényi contrastive learning and data augmentation [Reviewer B1D9]
 - Clearer motivation for using Rényi divergence (Section 1, Section 2) [Reviewer B1D9, Reviewer 6o5u]
 - Simpler notation for the definition of positive and negative pairs (Section 2) [Reviewer B1D9]
 - Added detailed analysis on Theorem 3.2 for constants (Section 3) [Reviewer UaDs]
 - Added stop-gradient operator for gradient analysis (Section 4) [Reviewer UaDs]
 - Added gradient analysis for $\alpha>0$ with ablation studies (Section 4, Appendix B) [Reviewer UaDs]
 - Updated the results for 300 epoch results (Section 5)
 - Simplified mutual information estimation parts (Section 5)
 - Added detailed explanations for mutual information estimation such as Monte-Carlo implementations (Appendix C)
 - Clarified limitations of our work (Section 6) [Reviewer UaDs]
 - Added future works regarding various aspects of contrastive learning (Section 6) [Reviewer dj8j]
 - Added comparison with other self-supervised methods (Appendix B) [Reviewer 6o5u]
 - Added ablations on the robustness of RényiCL (Appendix B) [Reviewer UaDs]

These updates are temporarily highlighted in “blue” for your convenience to check.

We sincerely believe that RényiCL can be a useful addition to the NeurIPS community, due to the above updates helping us better deliver the effectiveness of our method.

Thank you very much!

Paper 10007 Authors.

---

### Meta-Review · Area_Chair_VX4o · 2022-09-04

**Recommendation:** Accept
**Confidence:** Certain

**Metareview:**

The reviewers reached a consensus that this paper is a nice addition to NeuRIPS. Please refers to the reviews and author's responses for reviewers' opinions on the strength and weakness of the paper.



**Award:**

No

---

### Decision · Program_Chairs · 2022-09-14

Accept